**Observationally constrained analysis of sulfur cycle in the marine atmosphere with NASA**
**ATom measurements and AeroCom model simulations**
Huisheng Bian[1,2], Mian Chin[2], Peter R. Colarco[2], Eric C. Apel[3], Donald R. Blake[4], Karl Froyd[5], Rebecca S.
Hornbrook[3], Jose Jimenez[5,6], Pedro Campuzano Jost[5,6], Michael Lawler[5,7], Mingxu Liu[8], Marianne Tronstad Lund[9],
Hitoshi Matsui[8], Benjamin A. Nault[5,6,10,11], Joyce E. Penner[12], Andrew W. Rollins[5,13], Gregory Schill[7], Ragnhild B.
Skeie[9], Hailong Wang[14], Lu Xu[15,16], Kai Zhang[14], and Jialei Zhu[17]
[1]Goddard Earth Sciences Technology and Research (GESTAR) II, University of Maryland at Baltimore County,
Baltimore, MD, USA.
[2]NASA Goddard Space Flight Center, Greenbelt, MD, USA.
[3]Atmospheric Chemistry Observations & Modeling Laboratory, National Center for Atmospheric Research,
Boulder, CO, USA.
[4]Department of Chemistry, University of California Irvine, CA, USA.
[5]Cooperative Institute for Research in Environmental Sciences, University of Colorado, Boulder, CO, USA.
[6]Department of Chemistry, University of Colorado, Boulder, CO, USA.
[7]NOAA Chemical Sciences Laboratory, Boulder, CO, USA.
[8]Graduate School of Environmental Studies, Nagoya University, Nagoya, Japan.
[9]CICERO Center for International Climate Research, Oslo, Norway.
[10]Now at: Department of Environmental Health and Engineering, Whiting School of Engineering, The Johns
Hopkins, Baltimore, MD, USA.
[11]Now at: Center for Aerosol and Cloud Chemistry, Aerodyne Research, Inc., Billerica, MA, USA
[12]Dept. of Atmospheric, Oceanic and Space Sciences, University of Michigan, Ann Arbor, Michigan, USA.
[13]NOAA Earth System Research Laboratory, Chemical Sciences Division, Boulder, CO, USA.
[14]Atmospheric Sciences and Global Change Division, Pacific Northwest National Laboratory, Richland, WA, USA.
[15]Division of Geological and Planetary Sciences, California Institute of Technology, Pasadena, CA, USA.
[16]Now at Department of Energy, Environmental and Chemical Engineering, Washington University in St. Louis,
Missouri, USA.
[17]Institute of Surface-Earth System Science, School of Earth System Science, Tianjin University, Tianjin, China.
*Correspondence to*: Huisheng Bian (huisheng.bian@nasa.gov)
**Abstract**
The sulfur cycle plays a key role in atmospheric air quality, climate, and ecosystems, such as
pollution, radiative forcing, new particle formation, and acid rain. In this study, we compare the
spatial and temporal distribution of four sulfur-containing species, dimethyl sulfide (DMS),
sulfur dioxide ($SO_2$), particulate methanesulfonate (MSA), and particulate sulfate ($SO_4$), that
were measured during the airborne NASA Atmospheric Tomography (ATom) mission and
simulated by five AeroCom-III models to analyze the budget of sulfur cycle from the models.
This study focuses on remote regions over the Pacific, Atlantic, and Southern Oceans from near
the ocean surface to ~12-km altitude range, and covers all four seasons. These regions provide us
with highly heterogeneous natural and anthropogenic source environments, which is not usually
the case for traditional continental studies. We examine the vertical and seasonal variations of
these sulfur species over tropical, mid-, and high-latitude regions in both hemispheres. We
identify their origins from anthropogenic versus natural sources with sensitivity studies by
applying tagged tracers in GEOS model linking to emission types of anthropogenic, biomass
burning, volcanic, and oceanic emissions. Our work presents the first assessment of AeroCom
sulfur study using ATom measurements, providing directions for improving sulfate simulations,
which remain the largest uncertainty in radiative forcing estimates in aerosol climate models. In
general, the differences among model results can be greater than one-order of magnitude.
Comparing with observations, simulated $SO_2$ is generally low while $SO_4$ is high. Using

interactive oxidant calculation is insufficient to account for model sulfate bias. There are much larger DMS concentrations simulated close to the sea surface than observed, indicating that the DMS emissions may be too high from all models. The parameterization of converting DMS seawater concentrations into DMS emission fluxes needs to be revisited. Anthropogenic emissions are the dominant source (40-60% of the total amount) for atmospheric sulfate simulated at locations and times along the ATom flight tracks at almost every altitude, followed by volcanic emissions (18-32%) and oceanic sources (16-32%). Similar source contributions can also be derived at broad ocean basin and monthly scales, indicating that any reductions of anthropogenic sulfur emissions would have global impacts in modern times.

## 1. Introduction

Atmospheric sulfur species have wide-ranging environmental and health impacts. About two-third of sulfur emissions come from anthropogenic activities (Chin et al., 2000); therefore, considerable efforts have been made to reduce these sulfur emissions. For example, acid rain occurs when sulfur dioxide ($SO_2$) is oxidized to form sulfuric acid and particulate sulfate ($SO_4$), which fall to the ground with the rain (Bian et al., 1993; Grennfelt et al., 2020) and can devastate aquatic ecosystems (Josephson et al., 2014; McDonnell et al., 2021). Through the competing neutralization reaction of $SO_4$ and nitrate with $NH_3$ and other alkaline species, $SO_4$ affects strongly both particulate nitrate formation (Bian et al., 2017) and aerosol pH (Huang et al., 2020; Nault et al., 2021). Sulfate is a key component of particulate matter (PM), which degrades air quality (Dong et al., 2018; Tan et al., 2018) and directly reflects the sun's rays (Moch et al., 2022; Myhre et al., 2013). Due to its highly hygroscopic nature, sulfate aerosol affects cloud physics (Boucher et al., 2013; Breen et al., 2021; Seinfeld et al., 2016) and thus indirectly radiative forcing (Penner et al., 2016; Wang et al., 2021) through aerosol-cloud interactions. The contribution of aerosols to atmospheric clouds and energy budget remains the largest uncertainty in climate models (Gryspeerdt et al., 2023; Jia et al., 2021, 2022; Klein et al., 2013; Malavelle et al., 2017). Sulfate is important primarily because the atmospheric sulfate component itself contributes to radiation forcing (RF) almost as much as all other major non-natural aerosol components, as concluded from 16 AeroCom model results (Myhre et al., 2013). More importantly, uncertainty in sulfate simulations in current climate models is a major contributor to biases in aerosol optical depth (AOD, Fig. 3 in Gliß et al., 2021) and RF (Fig. 7 in Myhre et al., 2013).

Unlike other major atmospheric aerosols, a significant fraction (i.e., roughly a quarter) of sulfate in the atmosphere comes from marine biological emissions (Chin et al., 1996). The impact of oceanic sulfate is particularly pronounced on marine shallow clouds, which are characterized by low droplet number concentrations and weak updraft velocities (Rissman et al., 2004). Sulfur research has also focused on the tropical upper troposphere (TUT), where the growth of new aerosol particles and homogeneous nucleation involving sulfuric acid is at a maximum (Williamson et al., 2019), and where deep convective transport allows a small portion of the sources to reach the lower stratosphere. The resulting sulfate aerosols in the stratosphere can persist for years (Holton et al., 1995). Unfortunately, the observations in the TUT region and above are sparse. Acquiring atmospheric composition and its chemical/physical properties over remote oceans is challenging, although satellites can often provide total column constraints of aerosol optical depth.

The NASA Earth Venture Suborbital (EVS-2) Atmospheric Tomography (ATom) airborne
mission provided abundant measurements of gases and aerosols over the world's oceans (Hodzic
et al., 2020; Thompson et al., 2021). In particular, a suite of instruments integrated on the NASA
Douglas DC-8 jetliner (hereafter DC-8) made measurements of many important sulfur species
including dimethyl sulfide (DMS), $SO_2$, particulate methanesulfonate (MSA) and $SO_4$ over the
Pacific and Atlantic Oceans in both hemispheres and the Southern Ocean in all four seasons.
This comprehensive sulfur dataset provides us with unprecedented opportunities to assess sulfur
source, transport, chemistry, deposition, and particle activation and growth represented in the
global aerosol models, and to estimate the extent of anthropogenic influence on remote oceanic
atmospheric composition and cloud properties.
This study has two specific scientific goals. First, we explore the vertical and seasonal variation
of sulfur species (i.e., DMS, $SO_2$, MSA, and $SO_4$) using ATom measurements and simulations
from five global models that participated in the AeroCom-ATom model experiments. AeroCom
is an international initiative of scientists aiming at the advancement of the understanding of the
global aerosol and its impact on climate (https://aerocom.met.no/). Here we focus on remote
regions over the Pacific, Atlantic, and Southern Oceans, from near the surface to an altitude of
about 12 km, covering all four seasons. Second, we determine whether the produced $SO_4$
originated from anthropogenic or natural sources by using tagged tracers associated with
emission types.
Our work is the first study to use ATom measurements for comparison with the AeroCom
models, focusing on all sulfur species simulated in current aerosol climate models. This work
extends previous efforts using ATom measurements to evaluate the organic carbon (Hodzic et
al., 2020) and black carbon (Katich et al., 2018) of AeroCom models, as well as individual
models focusing on new particle formation in the tropics (Williamson et al., 2019), fine aerosol
lifetime (Gao al. al., 2022), aerosol vertical transport (Yu et al., 2019), sea salt (Bian et al.,
2019), smoke (Schill et al., 2020), mineral dust (Froyd et al., 2022), and DMS chemistry (Fung
et al., 2022). Furthermore, to our knowledge, there are no studies that systematically investigate
the changes and sources of all major sulfur species in the ocean. Our study aims not only to
reveal sulfur variability based on multiple measurements and model simulations, but also to tease
out the underlying processes behind the variability through a comprehensive analysis of
simulated sulfur species in aerosol climate models.
The structure of this paper is as follows. Section 2 describes the ATom measurements and the
AeroCom models used in this study. Section 3 presents the ATom-AeroCom sulfur comparison
from different perspectives, namely the overall comparison in Sect. 3.1, the vertical profiles in
Sect. 3.2, and the regional and seasonal analysis in Sect. 3.3. The sulfur budget analysis is given
in Sect. 4. We further present investigations of source origins for aerosol $SO_4$ along flight tracks
and over oceans in Sect. 5. Finally, we summarize our findings in Sect. 6.
**2. Data**
**2.1 ATom measurements**
ATom was a NASA-funded Earth Venture Suborbital project designed to study the effects of air
pollution on chemically reactive gases, aerosols, and greenhouse gases in the remote atmosphere.
ATom deployed a large suite of gas and aerosol measurement instruments on the NASA DC-8
aircraft for systematic sampling, covering an extended region of the globe from 85°N to 85°S
over the Pacific and Atlantic Oceans, with vertical profiles from near-surface to near-tropopause
(i.e., 0.2-12 km, Thompson et al., 2021). Four ATom deployments (ATom-1 to -4) were
executed over each of the four seasons from 2016 to 2018, and their flight paths are shown in
Fig. 1. The extensive aerosol and gas measurements made during ATom include inorganic and
organic aerosols, precursor gases, particle size distributions and particle composition. Table 1
lists the instruments for ATom sulfur species observations used in this study including the
relevant sampling details needed for the model comparison.
We use $SO_4$ and MSA that had been measured by two instruments, the University of Colorado
Aerodyne high-resolution time-of-flight aerosol mass spectrometer (AMS, Canagaratna et al.,
2007; Guo et al., 2021), and the NOAA Particle Analysis by Laser Mass Spectrometry (PALMS,
Froyd et al., 2019). The latter makes *in situ* measurements of the chemical composition of
individual aerosol particles. Furthermore, AMS measured submicron aerosols while PALMS
provided mass mixing ratio and size distribution up to 3 μm in dry diameter (Brock et al., 2019).
It is worth noting that AMS data were independently processed and reported at both 1-s and 60-s
time resolutions by instrument PI (Jimenez et al., 2019). The detection limit varied with different
averaging time resolutions, and they were provided directly for each sampling point in AMS
datasets. Some negative measurements were also presented in AMS datasets, and this is normal
for measurements of very low concentrations in the presence of instrumental noise. The AMS
data at 60-s resolution is recommended owing to more robust peak fitting at low concentrations
(Hodzic et al., 2020). Given the complex data overlays (i.e., starting, ending, and frequency)
reported from multiple instruments, the ATom team also provide a 10-s merged dataset to
facilitate users' applications. In this study, we evaluate data reported in different time
resolutions, using AMS as an example, to ensure the quality of merged data that are exclusively
used as the primary dataset in this work.
Two instruments were used for $SO_2$ measurements: the California Institute of Technology
Chemical Ionization Mass Spectrometer (CIMS) and the NOAA Laser Induced Fluorescence
(LIF) (Table 1). The CIMS uses $CF_3O^-$ as a reagent ion which reacts with $SO_2$ via fluoride ion
transfer chemistry. The product ion is detected by a compact time-of-flight mass spectrometer
(CToF). The precision of the CIMS $SO_2$ measurement decreases with increasing water vapor
concentration (Eger et al., 2019; Huey et al., 2004; Jurkat et al., 2016; Rickly et al., 2021),
making it challenging to measure $SO_2$ in remote ocean regions. In these regions, the ambient
water vapor may be sufficiently high that the CIMS $SO_2$ precision at 1-s resolution (~130 parts
per trillion by volume, pptv) is insufficient for measuring ambient $SO_2$ value there (<100 pptv).
To address this shortcoming, the ATom science team added a new instrument, the NOAA LIF, to
the ATom-4 payload. The NOAA LIF instrument uses red-shifted laser-induced fluorescence to
detect $SO_2$ at very low ppt levels (Rickly et al., 2021; Rollins et al., 2016). Both instruments
report negative values and the detection limit of the LIF instrument is about 2 pptv.
DMS was measured during ATom by two instruments, the University of California, Irvine
Whole Air Sampler (WAS), and the NCAR Trace Organic Gas Analyzer (TOGA). The WAS
reported DMS for all four ATom deployments, while the TOGA reported data for ATom-2 to -4
and not for ATom-1 due to possible issues associated with the TOGA inlet (the inlet was
changed for ATom-2 to -4). Both instruments have comparable detection limit (1 pptv) and
accuracy (~15%). However, the sampling time interval of WAS (variable but ~180s) was longer
than TOGA (~120s).

**2.2 AeroCom models**

Five global aerosol models participated in an AeroCom-ATom model experiment
(https://wiki.met.no/aerocom/phase3-experiments): CAM-ATRAS, E3SM, GEOS, IMPACT,
and OsloCTM3. The experiment required all participating models to (1) conduct three-year-
simulations of 2016-2018 (i.e., covering the whole ATom observation period); (2) use or nudge
meteorological data for the simulation period; and (3) use the same pre-defined emission fields
for precursor gases and aerosol tracers. The suggested emissions are the Coupled Model
Intercomparison Project Phase 6 Community Emissions Data System (CEDS, Hoesly et al.,
2018) for anthropogenic source, daily biomass burning emission (such as The Global Fire
Assimilation System, GFAS), a dataset based on satellite volcanic $SO_2$ observations from the
OMI instrument on the Aura satellite (Carn et al., 2016, 2017) for outgassing and eruptive
volcanic emission, and DMS concentration in sea surface from Lana et al. (2011). Wind-driven
emissions, such as dust and sea salt, are calculated online by each model. Table 2 summarizes
the detailed model characteristics and input datasets relevant to this study. It is worth noting that
CEDS specifies anthropogenic emissions from various sectors, including emissions from
shipping. The version of CEDS used in this work has emissions up to 2014 and all models use
2014 emission for ATom periods. Furthermore, unlike other models that use CEDS emissions,
the anthropogenic emissions of OsloCTM3 are obtained following Shared Socioeconomic
Pathways (SSP) under Representative Concentration Pathway (RCP) scenario with medium
radiative forcing by the end of the century (SSP245, Fricko et al., 2017), and the emissions are
interpolated to 2016 and 2017. Following the experimental protocol, all models provided results
for all AToms except for OsloCTM3 that omitted data in ATom-4. Unlike traditional AeroCom
experiments that used gridded daily/monthly averaged data, modelers are required to interpolate
model results along flight track every 10 s (see more discussion in Sect. 3.1) using three-
dimensional high frequency (e.g., hourly or even less depending on the models' time step) data
to facilitate the comparison. It is worth noting that the models do not have any actual information
at 10-s time resolution, given their time steps are at least 10× greater and their spatial resolutions
are coarse. However, the interpolation methodology suggested here provides the best model
information at their current configuration to compare with aircraft measurements.

The AeroCom-ATom experiment also designed three sensitivity simulations by tracking gas and
aerosol emissions to anthropogenic, biomass burning, and volcanic sources to attribute the origin
of sulfur sources on sulfur simulations over remote oceans. These experiments were conducted
with the Goddard Earth Observing System (GEOS) model. The setup of the GEOS model
followed the experiment protocol generally, but GEOS used its own daily biomass burning
emissions that were derived from the Quick Fire Emissions Dataset (QFED) developed based on
MODIS fire radiative power and calculated in near real-time at 0.1° resolution (Darmenov and
da Silva, 2015; Pan et al., 2020). Emissions from biogenic sources were calculated using the
Model for Emissions of Gases and Aerosols from Nature (MEGAN) embedded in the GEOS
model.

**2.3 Tag-tracer study in GEOS**

Tag trackers or tags are tied to sources of selected emission types and/or emission locations.
Such tag isolates plume from certain activities and is a powerful tool to help understand source
attribution or diagnose model performance at the process level. The mechanism behind this
technique is that each specific aerosol component in GEOS GOCART is modeled independently
of the other components, and the contribution of each emission type to the total aerosol mass is
not disturbed by the other emission types. Therefore, additional aerosol tracers can be easily
"tagged" to capture emission type (e.g., anthropogenic, biomass burning, etc.) and location
(local, regional or global scale). Tags can be multi-instantiated and computed simultaneously
with their baseline counterparts, thereby increasing the computational efficiency of scientific
research.
Tag-tracer technique in GEOS has been widely used in aerosol and gas studies (Bian et al., 2021;
Nielsen et al., 2017; Strode et al., 2018) and in supporting various aircraft field campaigns such
as Arctic Research of the Composition of the Troposphere from Aircraft and Satellites
(ARCTAS) and ATom. Such techniques are also adopted in other models such as GEOS-Chem
model (Fisher et al., 2017; Ikeda et al., 2017; Lin et al., 2020) and Community Earth System
Model (CESM, Butler et al., 2018).
Four tags linked to emission types of anthropogenic, biomass burning, volcanic, and marine
emissions were used in GEOS model to identify anthropogenic versus natural sources of sulfate,
and the results are discussed in Sect. 5.
**3. ATom-AeroCom comparisons of sulfur species**
This section presents a comparison of sulfur species between ATom measurements and
AeroCom model simulations. The consistency and diversity of data across remote regimes, both
horizontally and vertically, help us understand the effects of emissions, transport, and chemical
transformations, and shed light on improving the processes in models to best represent the ATom
observations.
**3.1 Overall comparison**
The overall performance of $SO_4$ PDF distribution observed from the AMS and PALMS
instruments and simulated by five AeroCom models for four ATom deployments is presented in
Fig. 2. Also shown in Fig. 2 are the corresponding various percentiles, namely, 0[th] (minimum),
25[th], 50[th] (median), 75[th], and 100[th] (maximum), and the mean for statistical analyses. The ATom
team provided a 10-s merged dataset deliberately by integrating data from various instruments to
a unified temporal resolution. We use this 10-s merged data where observations above detection
limit (DL) throughout the main text unless otherwise stated. When multiple instruments
measured the target field, only points where all instrument measured above DL values were
included in analysis, as AMS 10-s in red and PALMS 10-s in grey in Fig. 2. All model results
were sampled mimicking flight observations (see Sect. 2.2), and only data with measurements
available were used in comparison. This approach ensures that model evaluation is based on
high-quality measurements. It is worth noting that the given statistical values in this method
represent more regions having high tracer concentration or mixing ratio. In the supplementary
material, we further give a model-observation comparison for all available measurement data
including negatives.

The mean of PALMS $SO_4$ is generally about 10-50% higher than AMS $SO_4$ across four ATom
deployments. This performance may be attributed, at least in part, to the fact that the sample size
range of PALMS (~3 μm) is larger than that of AMS (~0.75 μm), as mentioned in Sect. 2.1.
However, the difference between the two observations is much smaller than the difference
between observation and model. Clearly, the differences in simulated $SO_4$ among models are
high and can easily exceed several orders of magnitude. Most observed and simulated $SO_4$
exhibit highest probability density around $SO_4$ values of 10-100 ng $sm^{-3}$. With the exception of
GEOS and CAM-ATRAS, the model $SO_4$ PDFs show higher tails beyond 100 ng $sm^{-3}$, which
explain the higher median and mean $SO_4$ simulated by the models. Statistical analysis performed
on selected percentiles (box-and-whisker panels in Fig. 2) indicates that multi-model $SO_4$
medians are about 3.7 (ATom-1), 2.2 (ATom-2), 1.9 (ATom-3), and 1.2 (ATom-4) times higher
than observed. In general, nearly all measurements and models indicate that $SO_4$ concentrations
on a global ocean basis are highest during the Northern Hemisphere (NH) spring season (ATom-
4). Similar analysis was also performed on all (e.g., both positive and negative) measurement
data (Fig. S2), the median/mean values of observations are naturally smaller than those in Fig. 2
by 8-20%, but the PDF distributions are almost identical between the two treatments.
Figure 3 shows the PDF distribution and statistics for $SO_2$. All observed and simulated data were
reprocessed by including points above the detection limit (2 pptv) only. Both instruments (CIMS
and LIF) were deployed during ATom-4. Despite CIMS being less precise than LIF (Rollins et
al., 2016), both instruments agreed within 95% and CIMS measured $SO_2$ concentrations were
consistently 3-7% lower than LIF measurements. This difference is within the combined
uncertainties of the two measurements, but it suggests a systematic calibration difference that is
currently unresolved (Rickly et al., 2021). Meanwhile, the width of CIMS $SO_2$ PDF (measured at
half-height) is narrower in ATom-4 than ATom-3, because of improved measurement precision
in ATom-4. The CIMS resolution was improved in ATom-4, which enables a better separation of
$SO_2$ and formate-$H_2O$. The CIMS $SO_2$ PDF in ATom-4 is around 10 pptv and is more consistent
with LIF measurements and model simulations. In contrast, the distribution of $SO_2$ measured by
CIMS during ATom-1 to -3 is spread much wider than the models. Throughout ATom periods,
models, especially E3SM, GEOS, and OsloCTM3, show higher peak heights and narrower peak
widths. Statistics indicate lower model $SO_2$ medians than observed (box-and-whisker in Fig. 3),
especially during ATom-1. However, the model means are comparable or even higher than those
observed, indicating that the models simulate unobserved episode events. Consequently, the
simulated mean/median ratio is higher than the observed value. Among the four ATom
deployments, ATom-4 has much better model observation consistency. Figure S3 presents the
corresponding analysis, including the measured negative values. Compared to Fig. 3, the
observed median and mean values drop substantially (up to 50%), but the model statistics change
relatively small.
Atmospheric DMS observations are scarce, especially on a global scale. Thus, DMS
measurements by the two instruments (WAS and TOGA) during the four ATom deployments
provide an unprecedented opportunity to investigate biological DMS over global remote oceans
and evaluate model DMS simulations on spatial and temporal distributions. By excluding points
with measured values below detection limit (i.e., 1 pptv), the overall DMS comparison in Fig. 4
indicates TOGA has higher data peaks and probability densities when DMS ranges from 3-10
pptv. However, this does not appear to be consistent with the lower median and mean values of
TOGA, indicating a higher tail in the WAS DMS PDF. Likewise, although the peak of WAS
DMS PDF is significantly higher than all models from 3-10 pptv (~5-20 pptv for ATom-3), the
median and mean of the WAS DMS are lower, suggesting an even higher tail in model DMS
PDF. Overall, there is a big gap between the WAS and TOGA DMS measurements, and both are
surprisingly low compared to the models. Statistical analysis performed on selected percentiles
(the box-and-whisker) indicates that multi-model DMS medians are about 4.9 (ATom-1), 8.6
(ATom-2), 6.6 (ATom-3), and 7.7 (ATom-4) times higher than observed, while model GEOS has
a better performance (i.e., 1.2, 2.7, 2.3, and 2.8 correspondingly). Even though the model DMS
median is mostly higher than the observed value, the degree of overestimation is not as serious as
the mean value that can be more than tenfold, indicating a few points are simulated with
extraordinarily high DMS values. Based on what we know about DMS sources and sinks, these
very large simulated DMS appear most commonly in the boundary layer (BL). Indeed it is
confirmed in Fig 5 by looking at the ratios of DMS median values between model simulations
and observations. The analyses are performed on four vertical ranges (e.g., the entire vertical
column, the BL 0-1.5 km, the low-middle free troposphere 1.5-6 km, and the upper troposphere
6-12 km). The last column "MMM/MOM" refers to multi-model median to multi-observation
median. The high ratio stems mostly from the BL, above which the consistency is much better.
Meanwhile, the PDF distribution and statistics of the models agree better with the WAS
measurement than with the TOGA measurement. We should also acknowledge that this is a very
limited set of observations we used here, and that there are some longer-term DMS observations
near the surface that were used as input for the parameterization of DMS emissions. More DMS
observations near the ocean surface are needed to make a confident comparison.
**3.2 Vertical profiles**
Vertical profiles of ATom-1 to -4 for observed and modeled $SO_4$, $SO_2$, DMS, and MSA are
shown in Figs. 6-9, respectively, for five latitude bands (from the north to the south) and for both
the Pacific and Atlantic Ocean basins. Again, the profiles include equal amounts of data for each
measurement and model result. In other words, all comparisons show only available points
where the two observed values (i.e., AMS vs. PALMS for SO4 and MSA, CIMS vs. LIF for
SO2, and TOGA vs. WAS for DMS) are greater than their detection limits, and where the model
values are extracted.
The average and range of sulfur tracers for ATom-1 to -4 are shown in Figs. 6-9 and their
corresponding details in each ATom are further given in Figs. S5-8. As shown in Fig. 6, the $SO_4$
measured by the two instruments are close to each other and lie generally within the span of
modelled $SO_4$ throughout the ATom periods. The spread of modeled $SO_4$ concentrations is large,
easily exceeding an order of magnitude, especially in the upper troposphere. Despite the need for
improvements, the models are generally able to capture the shape of the $SO_4$ profile.
Specifically, CAM-ATRAS and GEOS have good $SO_4$ vertical gradients over the tropical and
NH oceans, but their $SO_4$ values are too low compared to measurements over the Southern
Hemisphere (SH) free troposphere. The $SO_4$ of IMPACT and OsloCTM3 decreases too slowly
with altitude, as shown by their overestimated $SO_4$ values at high altitudes globally. E3SM
performed $SO_4$ simulations among other models. However, the performance of these models'
$SO_4$ vertical profiles cannot simply be explained by the way the oxidant is applied, because
among the five models, CAM-ATRAS, IMPACT, and OsloCTM3 used interactive oxidant
calculations, while E3SM and GEOS used archived oxidant data (Table 2). The complexity of
the chemistry deserves more attention. Of the five models, OsloCTM3 and GEOS participated in
the multi-model OH assessment (Nicely et al., 2000) and OsloCTM3 had a shorter methane
lifetime (relative to OH) than GEOS.
Figure 7 shows generally lower modeled $SO_2$ volume mixing ratios compared to the CIMS
observations for most altitudes and latitude bins. The spread among modeled $SO_2$ values exceeds
an order of magnitude around the measured $SO_2$. $SO_2$ is better simulated by model IMPACT in
the NH and by models CAM-ATRAS and OsloCTM3 in the SH than other AeroCom models.
The tropical Pacific appears to be an interesting region, with all models except GEOS failing to
capture observed local $SO_2$ sources. Basically, the observed $SO_2$ is high at the surface, falls
rapidly in the BL, and then gradually decreases above the BL, except for ATom-1, during which
a second peak appears just above the BL (see Fig. S6 for the details of ATom-1 to -4 separately).
These observations indicate a strong local source for $SO_2$ in all seasons and a transport source in
the low free-troposphere NH summer (ATom-1). Like observations, the model GEOS provides a
local source for $SO_2$ at the surface, but it misses the plume above the BL in ATom-1, and its
vertical $SO_2$ convection is consistently too weak. Since only one flight was in ATom-1, more
observations are needed to confirm whether GEOS has been failing to catch the plume there
during the NH summer. All other models show lower $SO_2$ at the surface than in the lower free
troposphere, which is inconsistent with the observed profiles. Figure S6 also shows an excellent
agreement of $SO_2$ profiles measured by the CIMS and LIF during ATom-4 and models agree
with measurements better in ATom-4 as well.
DMS measurements fill in another piece of the puzzle for the atmospheric sulfur budget. As
shown in Fig. 8, all five AeroCom models generally overestimate DMS in the BL, particularly
for models CAM-ATRAS and OsloCTM3. This large bias close to the surface requires us to
revisit the DMS emissions employed in our models. Of the five models, DMS emissions of
E3SM, and IMPACT are derived directly from climate emission inventories, while the DMS
emission of the other three models are parameterized using monthly climatological DMS
concentrations in sea water and surface meteorologies (e.g. surface wind and temperature, see
details in Table 3). Specifically, the parameterization used to convert DMS seawater
concentrations into DMS emission fluxes was using Nightingale et al. (2000) in CAM-ATRAS
and OsloCTM3 and Liss and Merlivat (1986) in GEOS. The three models used two inventories
of monthly DMS seawater concentrations, Lana et al. (2011) for CAM-ATRAS and GEOS, and
Kettle and Andreae (2000) for OsloCTM3. It is worth noting that even the latest climatological
database by Lana et al. (2011) was constructed by compiling measurements before 2000, so the
potential long-term change of DMS emission caused by environment change could be missed
(Barford, 2013). Also, although the data used by Lana et al. (2011) is large (i.e., ~47,000
seawater concentration measurements), interpolation and extrapolation techniques were still
necessary in creating a global monthly climatological DMS emission. Gali et al. (2018) reported
updated oceanic DMS levels on a global scale using remote sensing satellite data. However,
much effort is still needed to accurately establish global rates of change in order to create global
DMS emissions for climate modeling. This parameterization of air-sea exchange is important
because CAM-ATRAS and OsloCTM3, using the same parameterization but different DMS
seawater concentrations, reported close emissions in Sect. 4. On the other hand, the DMS
emissions of CAM-ATRAS are almost twice as high as those of GEOS. This difference in
emissions results from different parameterizations in the two models, since both models read the
same DMS seawater concentration.
Meanwhile, the modeled DMS vertical gradient is generally steeper than the observed one (e.g.,
Fig. 8 A54N-90N), implying slower vertical transport or faster chemical conversion of DMS to
$SO_2$ in the model. The data collected from the AeroCom models did not provide us with enough
information to obtain the determinants. Currently, GEOS and OsloCTM3 account for two
products from the oxidation of DMS (i.e., $SO_2$ and MSA) but only GEOS output MSA results.
The other models consider DMS oxidation products only as $SO_2$. These chemical processes in
the model may also need to be revisited. Previous studies proposed other chemical reactions for
DMS loss in the atmosphere. For example, halogen chemistry represented 71% of the DMS loss
in the study of Hoffmann et al. (2016). Veres et al. (2020) estimated that about 30% of DMS in
the atmosphere was oxidized to a sulfur compound, hydroperoxymethyl thioformate (HPMTF),
reported only in ATom-4. To this end, the HPMTF serves as a new reservoir of oceanic sulfur
and its life cycle in the atmosphere is unknown. The new finding indicates that important
components of Earth's sulfur cycle are not yet been fully understood and urges us to reassess this
fundamental marine chemical cycle. However, including these chemical DMS losses further
reduces DMS above the surface, making DMS in the models even lower at high altitudes.
Of the five models, only GEOS reports MSA (Fig. 9). The GEOS MSA matches observations in
the lower troposphere. In the upper troposphere (UT), the GEOS MSA tends to decrease slowly
or even increase with altitude. These patterns do not agree with observations, and this
inconsistency can be explained at least partially by the MSA phase stages defined in the model
and observations. AMS and PALMS only measure the particle phase of MSA, but GEOS MSA
is the total MSA and is not accurately represented by observations, especially in UT. Yan et al.
(2019) reported that the ratio of MSA to $SO_4$ can be reduced by 30% when calculations do not
consider methanesulfonic acid in the gas phase (MSAg) at low temperatures.
**3.3 Regional and seasonal analysis**
In order to analyze model performance on a regional and seasonal basis, Figs. 10-12 show
histograms of $SO_4$, $SO_2$, and DMS concentrations as a function of altitude (rows) and latitudinal
band (columns). Only multi-model median is shown here to highlight any common problems in
the models. Further details of each individual model are given in Figs. S9-11 and discussed in
supplementary material. Each model in this study has its anomalous behavior at a specific time
and location. With this knowledge, modelers can further explore the simulation to identify
potential causes of model anomalies.
High $SO_4$ concentration regions vary across seasons (Fig. 10). In the free troposphere (i.e., 1.5 –
12 km), these regions cover the tropics to mid-latitudes in summer and winter (i.e., ATom-1 and
ATom-2) and shift to mid- to high-latitudes in spring and autumn (i.e., ATom-3 and ATom-4).
The most high concentration areas appeared in the SH high-latitudes during ATom-3 (SH spring)
and the NH high-latitudes during ATom-4 (NH spring). Things are a bit more complicated in the
BL, but the tropical atmospheric $SO_4$ concentration appears to be always elevated, and $SO_4$
concentration levels and $SO_4$ interregional variation are more pronounced in ATom-1 (NH
summer). Among all AToms, the performance of the model $SO_4$ simulation is best for ATom-4
and worst for ATom-1 (NH summer). Compared to observations, model tends to simulate higher
$SO_4$ concentrations in the free tropospheric atmosphere. Both observations and simulations show
that the $SO_4$ in the Pacific is higher than that in the Atlantic during the NH high-latitude autumn
(ATom-3) and the NH mid-latitude spring (ATom-4). The differences between observations and
simulations are generally larger in the Atlantic than in the Pacific, particularly in the SH. $SO_4$
concentration levels in simulated and observed worlds can differ significantly in certain areas of
each ATom. Differences may be caused by majority models or a few individual models. For
example, in summer and winter, the CAM-ATRAS model gave the highest estimates of
atmospheric $SO_4$ in the oceanic BL, but the IMPACT and OsloCTM3 models gave the highest
estimates of atmospheric $SO_4$ in the free troposphere (Fig. S9). All models except the GEOS
model generally overestimate $SO_4$ in the atmosphere.
Atmospheric $SO_2$ (Fig. 11) is most abundant in the BL of NH mid-latitude Pacific Ocean during
ATom-1 (NH summer) and the tropical Pacific BL during ATom-3 (NH autumn), and this high
$SO_2$ region extends to the atmosphere above. Areas where free tropospheric $SO_2$ concentrations
are relatively large do not necessarily follow the example of the BL. For instance, free
troposphere appears to be more polluted than other regions in the NH Pacific during ATom-2
and in the SH mid-latitude Atlantic (A40S-20S) during ATom-4, but not in the BL, implying a
potential source of horizontal transport. The interregional variation of $SO_2$ in BL is much larger
than in the free troposphere, from which local oceanic sources of $SO_2$ can be inferred. In terms of
model-observation comparison, model simulated $SO_2$ in the free troposphere is generally lower,
which is opposite to the case of $SO_4$. A rapid $SO_2$ to $SO_4$ chemical conversion in models could
be one of reasons. Fig. S10 further shows individual model $SO_2$ simulation. For example, the
E3SM model gives significantly higher $SO_2$ compared with the measurements and other models
in BL (Fig. S10). Unlike the case of $SO_4$, all models tend to underestimate $SO_2$ in the free
troposphere, with some exceptions, such as the GEOS model for the mid- to high-latitude North
Pacific winter (ATom-2) and the CAM-ATRAS and IMAPCT models for the mid-latitude South
Atlantic autumn (ATom-4).
Surface DMS (Fig. 12) is generally higher in the tropics when the ocean is warm and in mid-high
latitudes during springtime (e.g., ATom-3 SH spring and ATom-4 NH spring). A remarkable
pattern of high model DMS values in the BL is revealed throughout the ATom cycle. This
phenomenon also occurs in the free lower troposphere, but not necessarily in the upper
troposphere. The high model DMS in BL can be attributed to (1) too high DMS emission, (2) too
slow DMS chemical loss, and (3) too slow DMS vertical transport from BL to free troposphere.
Additional insight can be obtained by focusing on remote high-latitudes, for example SH high-
latitude (40°S-70°S) Pacific, where land source impacts are limited. Thus, the higher simulated
$SO_2$ there in the BL in SYom-4 ruled out a chemical cause due to low DMS loss. The extremely
high surface DMS is also not due to the slow vertical transport because simulated DMS is also
high in the layers above the BL. A large model DMS emission is likely responsible for the
simulated high surface DMS. The overestimation of surface DMS multi-model median in Fig. 12
is clearly attributable to the contribution of all models shown in Fig. S11, with the models CAM-
ATRAS and OsloCTM3 being more prominent.
**4. Sulfur budget from AeroCom models**
Budget analysis is a simple and basic method that has been widely used to document the
underlying performance of a model. This analysis allows us to evaluate the AeroCom-III sulfur
simulations against previous AeroCom-I and -II studies and reserves a record for future model
evaluations. Table 4 summarizes the global sulfur budgets for emissions, wet/dry deposition and
chemistry from the five models. Clearly, the largest source of sulfur (~70 TgS/yr) is $SO_2$ emitted
directly from anthropogenic (~78%), biomass burning (~2%), and volcanic sources (~20%).
Biogenic DMS (~15-30 TgS) produced and outgassed from decomposition of marine organic
molecules provides the largest natural source of sulfur to the atmosphere. A small amount of $SO_4$
(< 3%) is emitted directly from anthropogenic sources.
DMS is oxidized in the atmosphere by OH and $NO_3$ radicals to form $SO_2$ and MSA. This
biological source of $SO_2$, along with $SO_2$ emitted directly from other sources, reacts with
hydroxyl radicals (OH) in the gas phase and hydrogen peroxide ($H_2O_2$) and ozone ($O_3$) in the
aqueous phase to produce sulfuric acid ($H_2SO_4$) and eventually sulfate particles, which play an
important role in the formation of clouds over the oceans.
In the five models, DMS has the shortest global average lifetime (0.6-2.0 days), followed by $SO_2$
(1.1-1.8 days), and $SO_4$ the longest lifetime (3.1-5.6 days). Among them, GEOS has the lowest
global burden and shortest lifetime for all sulfur species. The magnitudes of global burdens and
lifetimes shown here support the model performance shown in Figs. 2-8. For example, models
CAM-ATRAS and OsloCTM3 emit highest DMS, which is consistent with the highest DMS
value (Fig. 4 and S11) and longest lifetime simulated by the two models.
The key budget items include DMS emission, $SO_2$ emission, sulfate source or total deposition
(source and deposition are pretty much the same as expected), lifetime (reversely proportional to
the loss rate), and total atmospheric mass load. From the multi-model mean and standard
deviation, the "diversity" can be calculated. Figure 13 shows the global mean budget items in the
percentage deviation of each model from the multi-model mean, following the same concept
shown in Schulz et al. (2006) and Gliss et al. (2021). It reveals the processes causing model
differences. For example, E3SM and GEOS have approximately the same $SO_2$ emissions and
total sulfate sources, but the sulfate lifetime is much shorter in GEOS (implying faster removal
rates) thus smaller sulfate burden that is consistent with lower sulfate concentrations in GEOS
than in E3SM. At the same time, the lower total sulfate source in E3SM is compensated by
longer lifetime compared to CAM-ATRAS, resulting in a comparable global burden of $SO_4$ in
the two models.
It is worth pointing out that the much lower atmospheric $SO_4$ mass loading of the GEOS
simulations is not necessarily related to the poor performance of the GEOS $SO_4$ simulations, as
revealed by the model-measurement comparison in Figs 2, 6 and S9. Although the multi-model
mean (or median) often represents the best simulation in the modeling domain, common
modeling problems or too small model sample can compromise this effort.
To date, there have been no sulfur budget reports focusing on the vast ocean. However, previous
AeroCom studies have reported global sulfate atmospheric loading and its diversity across
multiple AeroCom models using monthly and global mean column loadings. Table 5 summarizes
these studies, including their reported global and annual sulfate multi-model mean (MMM) and
diversity ($\delta$). $\delta$ is related to the standard deviation (std_dev) and is defined as $\delta$ = std_dev /
MMM *100 (%). The results of this work are lower than AeroCom-I but higher than AeroCom-
II, which may be related to the different target years involved in these studies. One point to note
is that the diversity δ of AeroCom-III models has not reduced since AeroCom-I, which was
studied nearly 20 years ago.
**5. Source origins for aerosol SO$_4$ along flight track and Ocean basins**
In this section, we perform an analysis of source attribution by tagging the sulfur source types
using the GEOS model. This model is the only one that provides tagged data. Our goal is to
understand the sources (anthropogenic, biological, volcanic) of sulfate aerosols in remote regions
and how chemistry, transport, and removal processes determine the vertical distribution of
sulfate aerosols across seasons and ocean locations.
Figure 14a presents a quantitative summary of the source attribution of aerosol SO$_4$ sampled
along the ATom flight tracks. The analysis was performed over four seasons, spanning the
troposphere and three vertical layers (i.e., marine boundary layer, free troposphere and upper
troposphere). Overall, anthropogenic emissions were the dominant source (40–60% of the total)
of simulated tropospheric SO$_4$ along the ATom flight tracks for almost all altitudes and seasons,
followed by volcanic (18–32%) and oceanic sources (16–32%). Anthropogenic pollution
prevailed over remote oceans most in spring and autumn (ATom-3 and -4). The overall
contributions from volcanic and oceanic sources are comparable during the ATom periods.
Meanwhile, the ocean source contribution has an obvious seasonal variation which is most active
during the SH summer (ATom-2), when marine biochemical activity in the vast Southern Ocean
is the largest. Volcanos show the largest contribution in the NH summer 2016 (ATom-1) during
the four ATom deployments. Given the irregular and character of eruptions, the volcanic
contribution deserves further discussion below.
In the vertical direction, SO$_4$ from anthropogenic emissions contributes more than 50% to the
free to upper troposphere. Even in the marine boundary layer, anthropogenic sources of SO$_4$ still
account for the largest fraction, except in the SH summer (ATom-2) when oceanic source
became dominant. The relative importance of volcanic and marine sources varies not only
seasonally but also vertically. Oceanic sources understandably make up a significant fraction
(26-42%) of SO$_4$ in the boundary layer. In the free troposphere, its contribution drops off
sharply, reflecting its local surface source characteristics. On the other hand, SO$_4$ from
anthropogenic emissions (including shipping emission) expands in the free troposphere,
suggesting that the source originated from distant continental areas. Volcanic SO$_4$ remains nearly
constant throughout the troposphere, making volcanoes the second largest source there.
Meanwhile, the contribution of others (OTH including biomass burning) to remote ocean SO$_4$ is
relatively small (< 3%) and will not be discussed further in this study.
The sources of SO$_4$ discussed above are deduced from the location and timing of the ATom
flight path. Conclusions about the total contribution of the ocean needs caution, as there may be
representativeness issues using such narrow-band and instantaneous sampling. There might be a
situation where, for example, volcanoes provide a very large signal but only account for a small
measured area, and in most regions, volcanoes play a very minor role. Whereas oceanic sources
in the marine boundary layer perhaps were the dominant source for a much wider region but the
SO$_4$ concentration resulting from the DMS was overall a smaller amount compared to other
sources where near a volcanic or anthropogenic source. To address this representation issue, we
perform one more analysis with the model data averaged over a wider oceanic region (the shaded
orchid area in Fig. 1) and over a longer period (i.e., monthly mean over ATom periods). Such
source attributions are given in Fig. 14b.
Qualitative conclusions drawn from source attribution along the flight tracks generally apply to
the ocean basin source attribution, albeit to a slightly different extent. This confirms that
continental man-made sources dominate tropospheric $SO_4$ even over oceans. There is a clear
seasonal variation in oceanic contribution, which is largest in austral summer (ATom-2)
followed by boreal summer (ATom-1). Concerning volcanic sources, emissions from volcanoes
are of two types. One type is the volcanic degassing emissions that tend to remain nearly
constant throughout the year and are equivalent to about 20% of $SO_2$ global anthropogenic
emissions. This degassing emission ensures that volcanoes contribute more than 20% to $SO_4$
over the oceans. The other type consists in the volcanic eruptions. Due to the irregularity of
volcanic eruptions in terms of different eruption locations, magnitudes, and times, volcanic
eruptions can cause severe fluctuations in $SO_4$ in the atmosphere. Compared with the source
attribution along the flight trajectory, the volcanic contribution decreased over a larger spatial
and temporal domain (i.e., ocean basin and monthly mean) in the NH winter 2017 by 32%
(ATom-2) and increased in all other three seasons by 14-33%, especially in the NH spring 2018
(ATom-4), when the massive Kilauea eruption in Hawaii began on 3 May 2018. Contrarily, the
anthropogenic contribution increased in the NH winter (ATom-2) by 5% and decreased in other
seasons by 7-21%.
**6. Conclusions**
This study investigates sulfur in remote tropospheric regions at global and seasonal scales using
airborne ATom measurements and AeroCom models. The goal is to understand the atmospheric
sulfur cycle over the remote oceans, each model's behavior and the spread of model simulations,
as well as the observation-model discrepancies. Such understanding and comparison with real
observations are crucial to narrow down the uncertainty in model sulfur simulation. Even after
decades of development, models are still struggling to accurately simulate sulfur distributions,
with differences between models often exceeding an order of magnitude. On the other hand, the
agreement between instruments is usually much better. Differences between modeled $SO_4$ are
particularly large in the tropical upper troposphere, where deep convective transport allows a
small portion of sulfur sources to reach the lower stratosphere where resultant sulfate aerosols
can persist for many years. Compared with observations, simulated $SO_2$ is generally low while
$SO_4$ is high. Modeled DMS values are typically an order of magnitude higher than observed
DMS near the surface, pointing to a need to revisit the DMS emission inventories and/or the
biogeochemical modules used to predict DMS emissions. Our work also suggests investigating
three other potential corresponding processes: whether the chemical conversion from $SO_2$ to $SO_4$
is too rapid, whether DMS-generated free tropospheric $SO_2$ is too low, and whether the vertical
transport of DMS and $SO_2$ from BL to free troposphere is too low. This further investigation
requires atmospheric oxidant fields and the ability to track $SO_2$ production and loss using tagged
tracers.
We investigate source attribution of $SO_4$ over remote oceans seasonally and vertically. Sampled
at the location and time of ATom measurements, anthropogenic emissions were the dominant
source (40–60% of the total) of simulated tropospheric $SO_4$ at almost all heights and seasons,
followed by volcanic (18–32%) and oceanic sources (16–32%). These contributions changed to
34–56%, 17–37%, and 19–37% when extended to the broad Pacific and Atlantic during the
months of ATom deployment. This survey confirms that anthropogenic sources dominate
tropospheric $SO_4$ even over oceans. Given that we find DMS source to be overestimated in the
models, the anthropogenic sources overall are a larger portion of the budget, and biogenic is
likely smaller than volcanic. Volcanic degassing throughout the year contributes about 20%, and
this proportion is increased by explosive eruptions that vary in location and timing. The oceanic
contribution has obvious seasonal variation, the largest in the Southern Hemisphere summer,
followed by the Northern Hemisphere summer.
It is understood that anthropogenic sulfur emissions currently offset a significant portion of
greenhouse gas warming, but they are rapidly declining through emissions controls. As these
anthropogenic emissions decrease, natural sources of sulfur, particularly bio-derived sulfur
compounds discharged from the world's oceans, will increase their relative contribution.
Therefore, more efforts are needed to understand the sulfur cycle in remote environments. On the
other hand, our study is the first asserting that anthropogenic emissions remain a major source of
sulfate aerosols generated over remote oceans during the ATom deployment periods, suggesting
that any limitation of anthropogenic sulfur emissions would have modern global implications.
Even after two decades of development, the diversity of sulfate simulations from AeroCom-I to
AeroCom-III has not decreased. However, accurate sulfate simulation in current climate models
is crucial to reduce radiative forcing biases. Several potential directions for improving sulfur
simulations are suggested above. More importantly, apart from the shortcomings of individual
models, all modelers should focus on the calculation of the air-sea exchange flux formula, as it
plays a key role in determining DMS emissions. Modelers also need to study DMS and $SO_2$
vertical transport as well as $SO_4$ wet deposition during long-distance transport, as model biases
are greatest at high altitudes. One suggestion to modelers is that the use of online oxidant fields
is insufficient to explain the model sulfate bias, as there was no systematic bias in the sulfate
simulations between the models using interactive oxidants and the models using archival
oxidants in this study. The complexity of chemistry deserves more attention.

*Code availability.* The GEOS Earth System Model source code and the instructions for model build are available
at https://github.com/GEOS-ESM/GEOSgcm/ (Last accessed: 28 August 2023).

*Data availability.* The AeroCom model outputs needed to reproduce the results described in this paper are
publicly available for download at https://acd-ext.gsfc.nasa.gov/anonftp/acd/tropo/bian/ATom-AeroCom-Sulfur/.
The ATom data was obtained from their ESPO Data Archive: https://espo.nasa.gov/atom/content/ATom, last
accessed: 28 August 2022.

*Author contributions.*
BH and MC conceptualized ATom-AeroCom experiment. BH performed analysis and wrote the manuscript. BH,
PRC, MLi, MTL, RBS, HM, JEP, HW, KZ, and JZ provided AeroCom model results and ECA, KF, RSH, JJ, PCJ,
MLa, BAN, AWR, GS, and LX contributed to ATom measurements. All authors contributed to the editing of the
manuscript.

*Competing interests.*
At least one of the co-authors is a member of the editorial board of Atmospheric Chemistry and Physics.

*Acknowledgements.*
HB, MC, and PRC acknowledge the GEOS model developmental efforts at Global Modeling and Assimilation Office (GMAO). This work was supported by NASA's Aura STM and ISFM programs and ACMAP award (80NSSC23K1000). The computing resources supporting this work were provided by the NASA High-End Computing (HEC) Program through the NASA Center for Climate Simulation (NCCS).
ECA and RSH acknowledge the support of the National Center for Atmospheric Research, which is a major facility sponsored by the National Science Foundation under Cooperative Agreement No. 1852977.
MLi acknowledges the support of JSPS Postdoctoral Fellowships for Research in Japan (Standard).
HM was supported by the Ministry of Education, Culture, Sports, Science, and Technology and the Japan Society for the Promotion of Science (MEXT/JSPS) KAKENHI grants (JP19H05699, JP19KK0265, JP20H00196, JP20H00638, JP22H03722, JP22F22092, JP23H00515, JP23H00523, and JP23K18519); by the MEXT Arctic Challenge for Sustainability II (ArCS II) project (JPMXD1420318865); and by the Environment Research and Technology Development Fund 2-2003 (JPMEERF20202003) and 2-2301 (JPMEERF20232001) of the Environmental Restoration and Conservation Agency.
KZ and HW acknowledge support by the U.S. Department of Energy (DOE), Office of Science, Office of Biological and Environmental Research, Earth and Environmental Systems Modeling program. The Pacific Northwest National Laboratory (PNNL) is operated for DOE by Battelle Memorial Institute under contract DE-AC05-76RLO1830.
LX thanks Michelle Kim, Hannah Allen, John Crounse, and Paul Wennberg for operating the Caltech CIMS instrument during ATom. LX acknowledges NASA grant NNX15AG61A.
MTL thanks Marit Sandstad (CICERO) for assistance with the model post-processing and acknowledges the National Infrastructure for High Performance Computing and Data Storage in Norway (UNINETT) resources (grant NN9188K).
RBS acknowledges funding from the Research Council of Norway (grant number 314997).

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

Table 1. ATom sulfur measurements used in the study

| | SO$_4$ | | SO$_2$ | | MSA | | DMS | |
|---|---|---|---|---|---|---|---|---|
| Instrument | AMS[a] | PALMS[b] | CIMS[c] | LIF[d] | AMS | PALMS | TOGA[e] | WAS[f] |
| ATom deployment(s) | 1 to 4 | 1 to 4 | 1 to 4 | 4 | 1 to 4 | 1 to 4 | 2 to 4 | 1 to 4 |
| Frequency | 60 s | 180 s | 1 s | 1 s | 1 s | 180 s | 120 s | Variable but ~180 s |
| Accuracy | ±35% (2s) | ±60% at 10 ng m-3 ±20% at 1 µg m-3 | ±25% | ± 9% (1s) | ±35% (2s) | ±70% | 15% or better | 15% |
| precision | | | 130pptv | | | | | 10% |
| Detection limit | 5-15 ng sm$^{-3}$ | ~10 ng sm$^{-3}$ | | 2 pptv | 2.5 ng sm-3 (60 s) | ~15 ng sm$^{-3}$ | 1 ppt | 1 ppt |
| Cut-off size (dry diameter) | ~0.75 µm | 0.1-3 µm | | | ~0.75 µm | 0.1-3 µm | | |
| Primary Investigator(s) | Jose Jimenez and Pedro Campuzano Jost | Karl Froyd and Gregory Schill | Paul Wennberg | Andrew Rollins | Jose Jimenez and Pedro Campuzano Jost | Karl Froyd and Gregory Schill | Eric Apel | Donald Blake |
| References | Guo et al., 2021; Schueneman et al., 2021 | Froyd et al., 2019 | Allen et al., 2022; Crounse et al., 2006 | Rollins et al., 2016 | Hodshire et al., 2019 | Froyd et al., 2019 | Apel et al., 2015 | Simpson et al., 2001 |

[a]AMS: Aerosol Mass Spectrometer
[b]PALMS: Particle Analysis by Laser Mass Spectrometry
[c]CIMS: Chemical Ionization Mass Spectrometer
[d]LIF: Laser Induced Fluorescence
[e]TOGA: NCAR Trace Organic Gas Analyzer
[f]WAS: Whole Air Sampler

Table 2. AeroCom Models used in this study

| Model Abbreviation | Model Version | Nominal Resolution | Vertical Levels | Meteorological Fields | Ocean Surface Temperature Data | Interactive Aerosol-Meteorology | Endogenous Oxidants | Endogeous DMS Emission | Aerosol Module | Anthropogenic Emission | Volcano Emission | Key References |
|---|---|---|---|---|---|---|---|---|---|---|---|---|
| CAM-ATRAS | CAM5-ATRAS2 | 1.9° × 2.5° | 30 | MERRA-2 | HadSST | Yes | Yes | No | Microphysics, 12 sectional size bins, and internal mixing of aerosol constituents in each bin. | CEDS (Hoesly et al., 2018), | Degassing (Andres and Kasgnoc, 1998), Eruption (Neely and Schmidt, 2016) | Liu and Matsui 2021; Matsui 2017; Matsui and Mahowald, 2017 |
| E3SM | v1.0 | 1° × 1° | 72 | ERA-Interim | HadSST | Yes | No | No | Microphysics, MAM4, internal mixing within a mode, external mixing between modes | CEDS (Hoesly et al., 2018) | Contineous emission (Denener et al., 2006). No eruptive emissions. | Rasch et al., 2019; Wang et al., 2020; Zhang et al. 2022 |
| GEOS | Icarus-3_3_p2 | 1° × 1° | 72 | MERRA-2 | MERRA sst | Yes | No | Yes | GOCART, Bulk, external mixing | CEDS (Hoesly et al., 2018) | Carns et al., 2016, 2017 | Bian 2017; Colarco et al., 2010; Chin et al., 2000 |
| IMPACT | | 1.9° × 2.5° | 30 | Open IFS ECMWF | HadSST | No | Yes | no | Microphysics, internal mixing within a mode, external mixing between modes | CEDS (Hoesly et al., 2018) | AeroCom volcanic emissions | Zhu et al., 2017; Zhu et al., 2019 |
| OsloCTM3 | OsloCTM3v1.02 | 2.25° × 2.25° | 60 | Open IFS ECMWF | Open IFS ECMWF | No | Yes | Yes | Bulk, external mixing | SSP245 with linear interpolation for 2017 | AeroCom volcanic emissions, continuous from Dentener (2006) | Lund et al., 018; Søvde et al., 2012 |

Table 3. DMS emission used/calcuated by the five AeroCom models

| Model abbreviation | Emission inventory | DMS concentration in sea water | DMS flux calculation | Meteorological fields |
|---|---|---|---|---|
| CAM-ATRAS | No | Lana et al. (2011) | Nightingale et al. 2000 | Wind from ECMWF-IFS |
| E3SM | Yes | | | |
| GEOS | No | Lana et al. (2011) | Liss and Merlivat, (1986), Saltzman et al. (1993) | SST and wind from GEOS |
| IMPACT | Yes | | | |
| OsloCTM3 | No | Kettle and Andreae (2000) | Nightingale et al. (2000) | Wind from ECMWF-IFS |

Table 4. Global sulfur budget in 2017

| | | Emission | SUPSO₂[1] | SUPMSA | SUPSO₄ | Dry | Wet | TotalSource | Burden | Lifetime |
|---|---|---|---|---|---|---|---|---|---|---|
| | | TgS/yr | TgS/yr | TgS/yr | TgS/yr | TgS/yr | TgS/yr | TgS/yr | TgS | days |
| CAM-ATRAS | DMS | 26.05 | -26.05 | -- | -- | -- | -- | 26.05 | 0.13 | 1.8 |
| | SO2 | 68.67 | 26.05 | -- | -55.67 | -39.05 | | 94.72 | 0.445 | 1.7 |
| | SO4 | 1.76 | -- | -- | 55.67 | -4.72 | -53.23 | 58.09 | 0.67 | 4.2 |
| E3SM | DMS | 19.43 | -19.40 | -- | -- | -- | -- | 19.43 | 0.0658 | 1.24 |
| | SO2 | 67.92 | 19.40 | -- | -38.56 | -48.76 | | 87.32 | 0.3825 | 1.60 |
| | SO4 | 1.74 | -- | -- | 38.56 | -6.95 | -33.31 | 40.31 | 0.6183 | 5.60 |
| GEOS | DMS | 15.57 | -14.84 | -0.74 | -- | -- | -- | 15.57 | 0.0252 | 0.59 |
| | SO2 | 67.06 | 14.84 | | -37.49 | -32.93 | -11.39 | 81.90 | 0.3488 | 1.55 |
| | SO4 | 1.68 | -- | -- | 37.49 | -5.27 | -33.90 | 39.17 | 0.3269 | 3.05 |
| | MSA | -- | -- | 0.74 | -- | -0.10 | -0.64 | -.74 | 0.0063 | 3.11 |
| IMPACT | DMS | 18.22 | -18.22 | -- | -- | -- | -- | 18.05 | 0.0369 | 0.75 |
| | SO2 | 64.76 | 18.22 | -- | -51.44 | -31.29 | -- | 82.98 | 0.4134 | 1.82 |
| | SO4 | 1.36 | -- | -- | 51.44 | -3.48 | -49.32 | 52.80 | 0.7502 | 5.19 |
| OsloCTM3 | DMS | 26.93 | -26.93 | -- | -- | -- | -- | 26.93 | 0.1496 | 2.03 |
| | SO2 | 52.80 | 26.93 | -- | -49.23 | -29.01 | -1.49 | 79.73 | 0.2346 | 1.08 |
| | SO4 | 1.053 | -- | -- | 55.49 | -6.35 | -50.29 | 56.54 | 0.8681 | 5.60 |

[1]SUPSO₂: chemical production for SO₂
Table 5. Global and annual sulfate multimodel mean and diversity from three AeroCom phases

| | AeroCom-I | AeroCom-II | | AeroCom-III | |
|---|---|---|---|---|---|
| reference | Textor et al., 2006 | Myher et al., 2013 | Kipling et al., 2016 | Gliß et al., 2021 | This work |
| Study year | 2000 | 2006 | 2006 | 2010 | 2017 |
| # of models | 16 | 16 | 18 | 14 | 5 |
| MMM (Tg) | 2.0 | 1.05 | 1.48 | 1.87 | 1.94 |
| δ (%) | 25.0 | 26.4 | 34.6 | 38.8 | 28.0 |
| observation | No | No | No | AC, AS, AE, and AOD from Ground station and AOD from MODIS | DMS, SO₂, SO₄ and MSA from ATom |


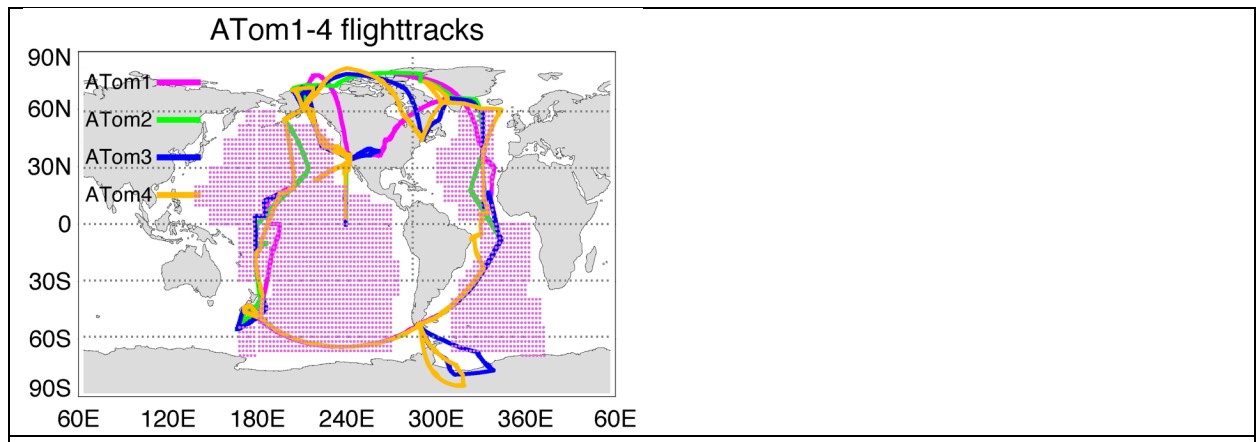

Figure 1. Flight tracks of ATom-1 to -4 and regions for the analysis of $SO_4$ source origins (shaded area). Periods of the four ATom deployments are ATom-1 (July-August 2016), ATom-2 (January-February 2017), ATom-3 (September-October 2017) and ATom-4 (April-May 2018).


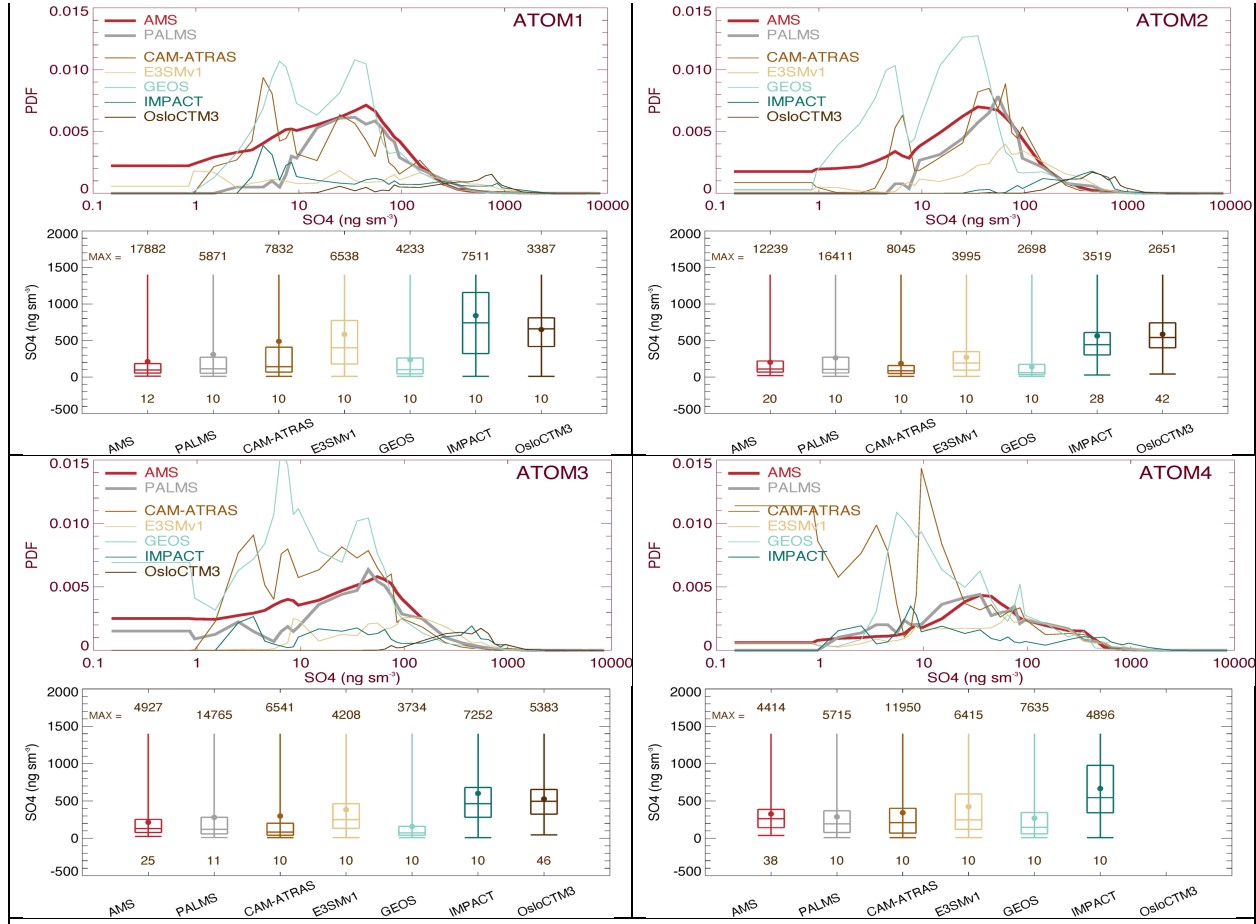

Figure 2. SO$_4$ probability density functions (PDF) and its statistical values shown by box-and-whisker for the four ATom deployments. All data (AMS in red, PALMS in grey, and five model simulations in other colors) are sampled at 10-s points. Statistical values are calculated when measured values are above the detection limit (DL). Statistical values include the range of the data from minimum to maximum, the three levels of the 25th, 50th (median), and 75th percentiles in the box, and the filled circle for the mean.


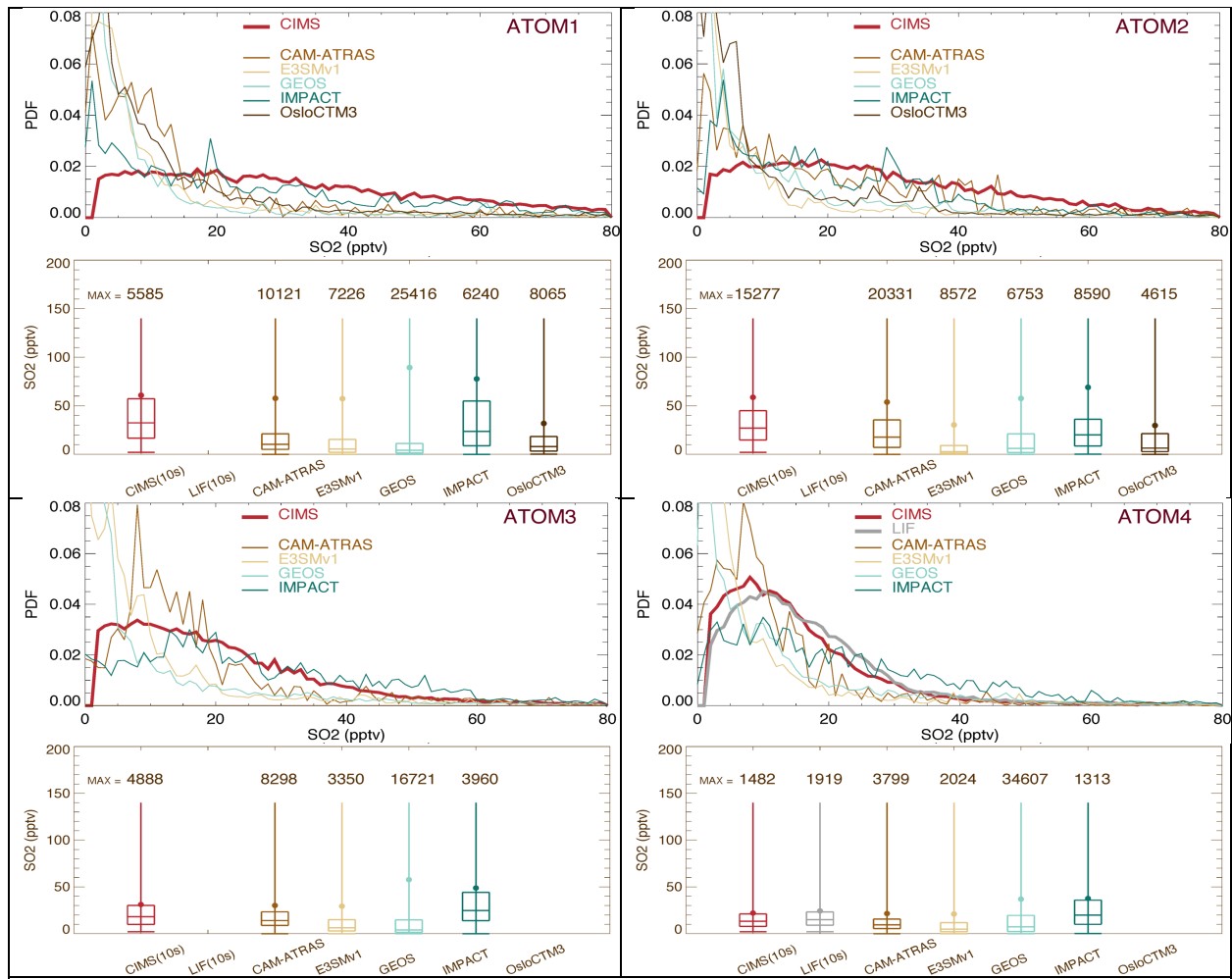

Figure 3. Similar to Fig. 2 but for SO₂. Observational data are CIMS (red) for ATom-1 to -4 and LIF (grey) for ATom-4 from ATom 10-s merged data. PDFs and statistical values are calculated at points where CIMS (and LIF in ATom-4) measured SO₂ are above DL (e.g., 2 pptv).


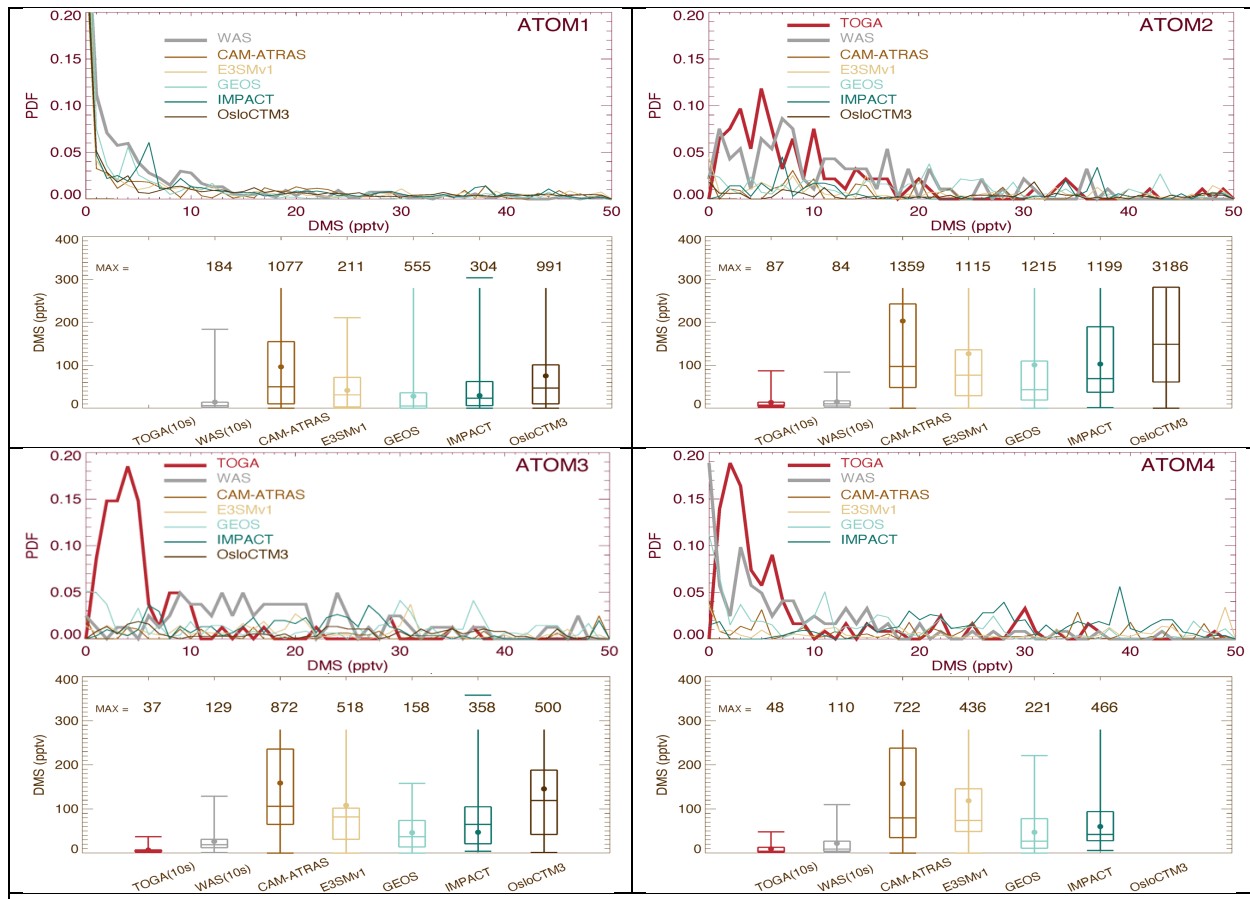

Figure 4. Similar to Fig. 2 but for DMS for ATom-1 to -4. The original data reported by TOGA (e.g., 35-s) and by WAS (e.g., ~60-s) have also been converted to 10-s frequency. Data included in PDF and statistical analysis are on 10-s points where DMS measured by both TOGA and WAS are above DL (i.e., 1 pptv).


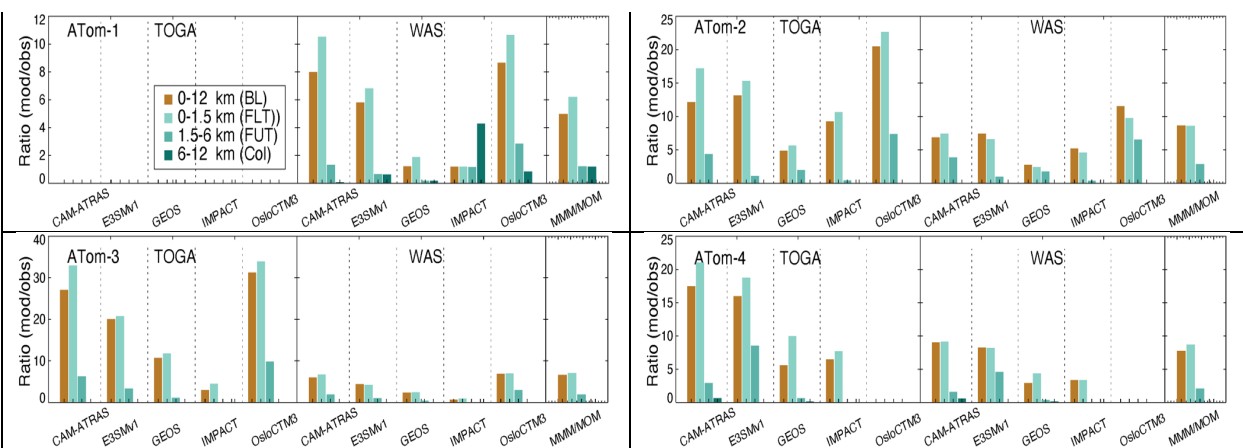

Figure 5. Ratio of DMS median values between model simulation and observation for four ATom deployments. Ratio analyses are performed on four vertical ranges as shown in four colors (see legend in ATom-1). The last column "MMM/MOM" refers to multi-model median to multi-observation median.

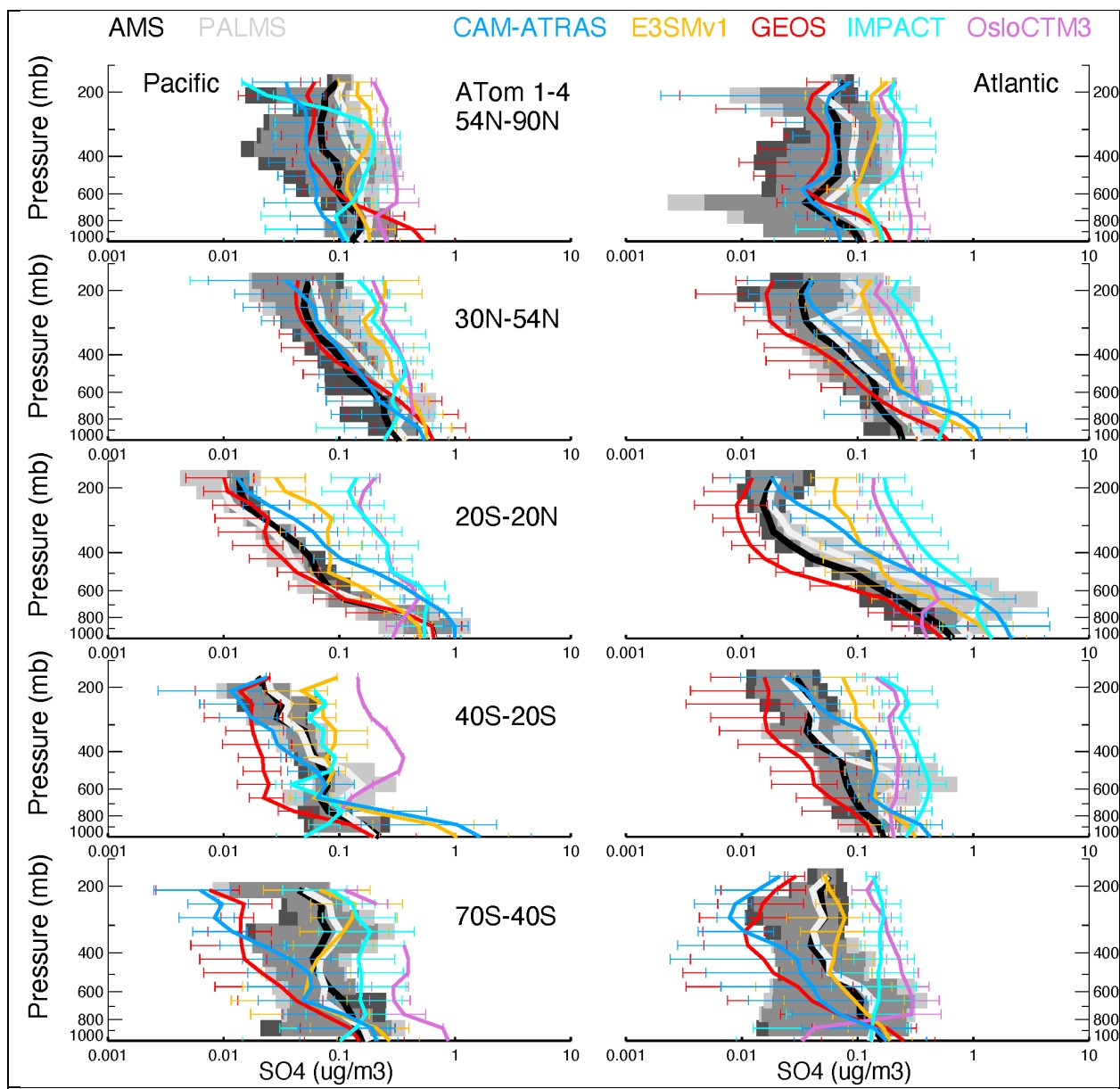

Figure 6. Observed and modeled vertical profiles of SO₄ in 1-km vertical bins averaged for four ATom deployments (lines) and variation across the four AToms (shaded area for measurements and horizontal bars for simulations). ATom measurements are shown in black (AMS) and light grey (PALMS) while model results are shown in other colors. Comparisons are conducted only when both observational measurements above detect limitation are available. Comparisons are separated into five latitude bands from the Northern to the Southern Hemisphere, and into Pacific and Atlantic Basins.


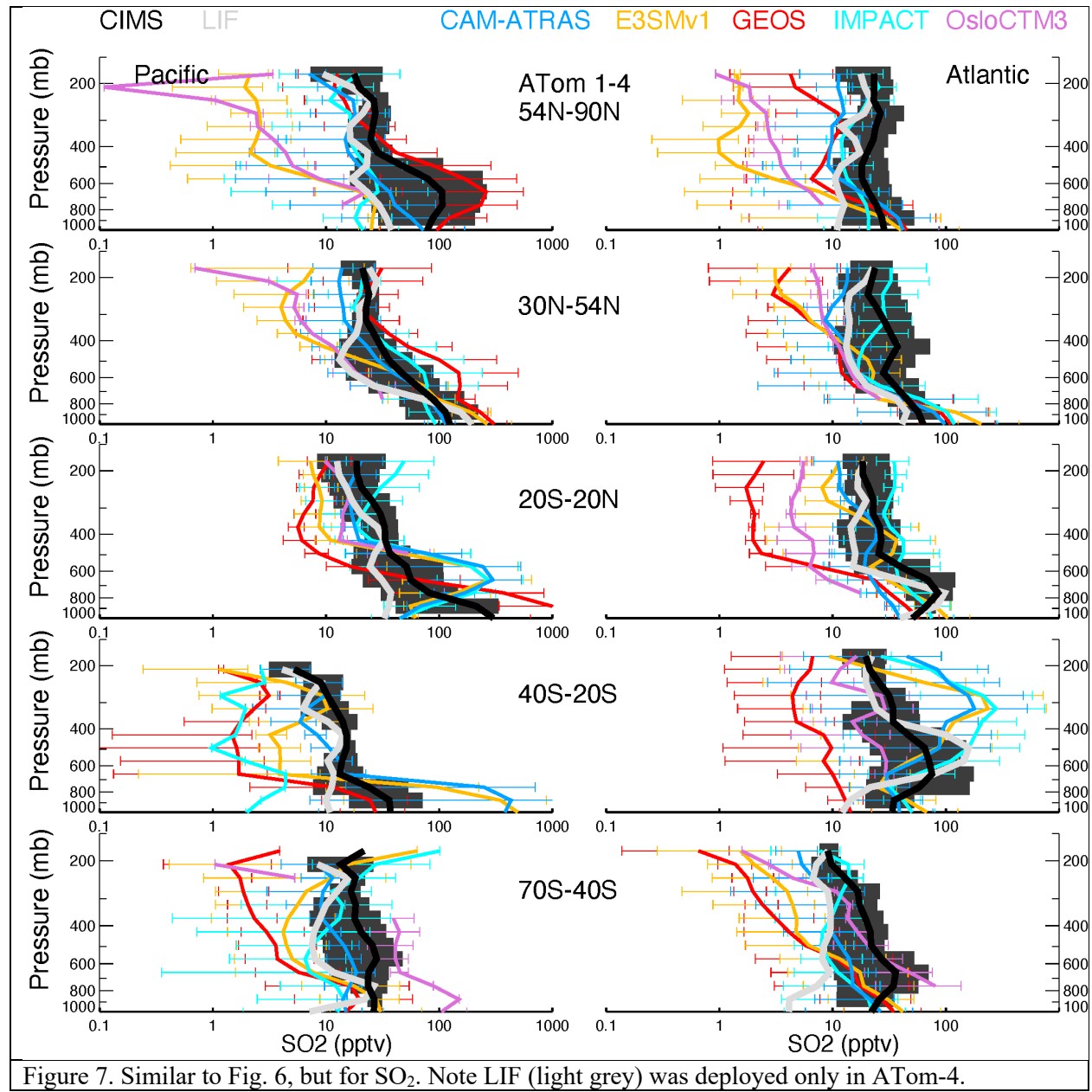

Figure 7. Similar to Fig. 6, but for SO₂. Note LIF (light grey) was deployed only in ATom-4.


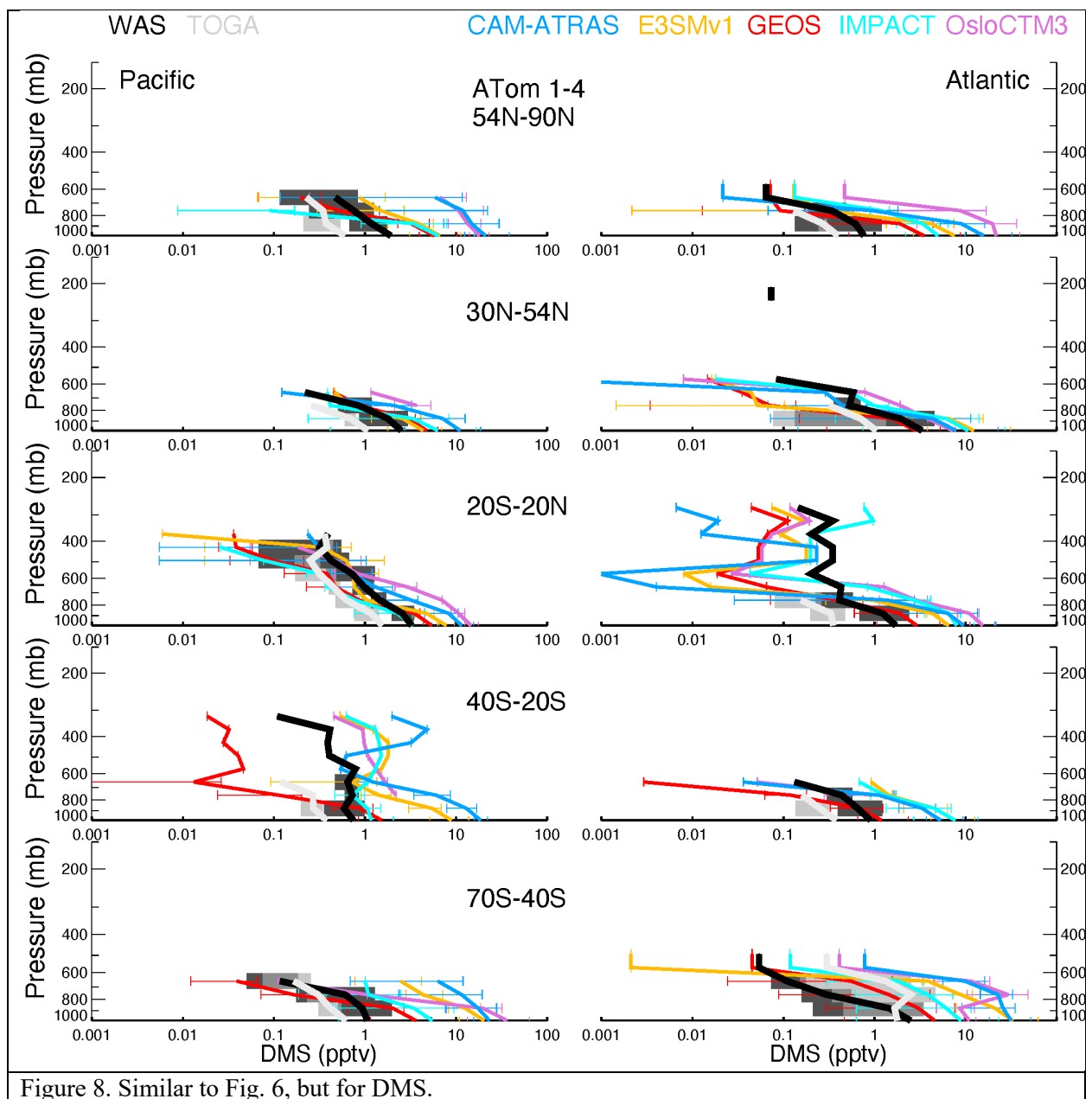

Figure 8. Similar to Fig. 6, but for DMS.


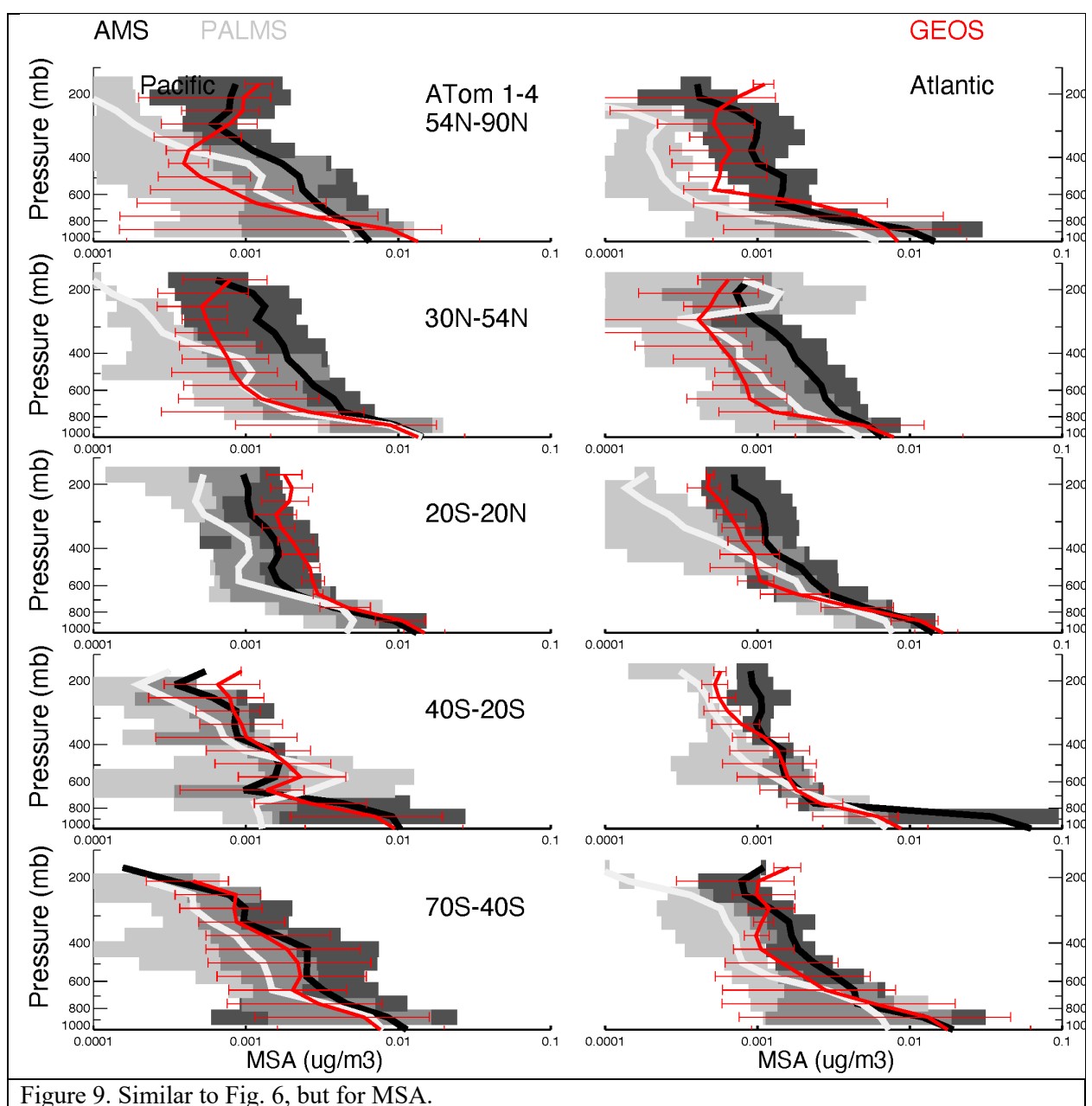

Figure 9. Similar to Fig. 6, but for MSA.


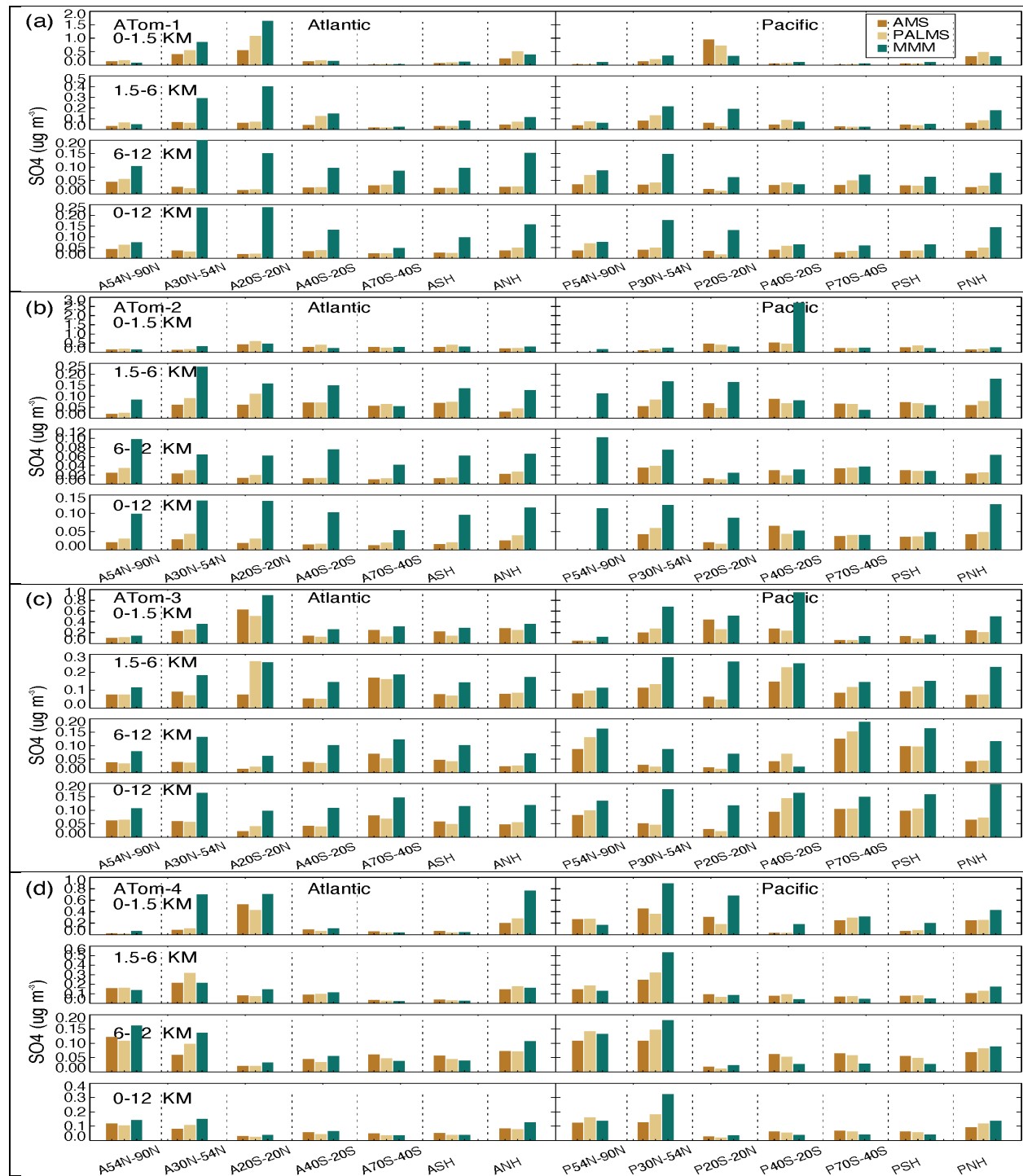

Figure 10. Median SO₄ concentrations from two measurements (AMS orange and PALMS yellow) and multi-model simulation (green) at seven latitudinal bands (including SH and NH) and four vertical layers (i.e., 0-1.5 km, 1.5-6 km, 6-12km, and 0-12 km) over Atlantic and Pacific oceans for four ATom deployments (a-d).


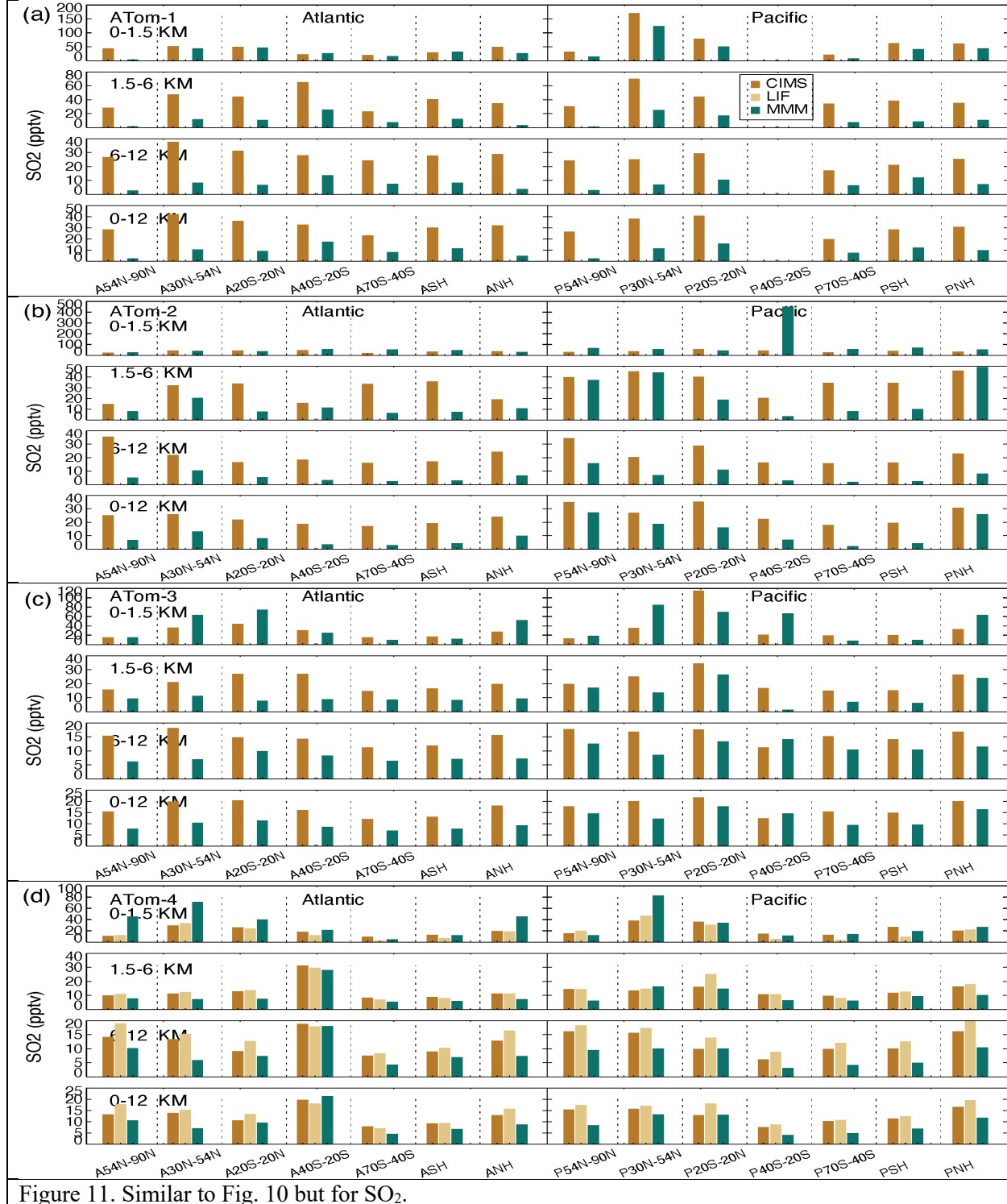

Figure 11. Similar to Fig. 10 but for SO₂.

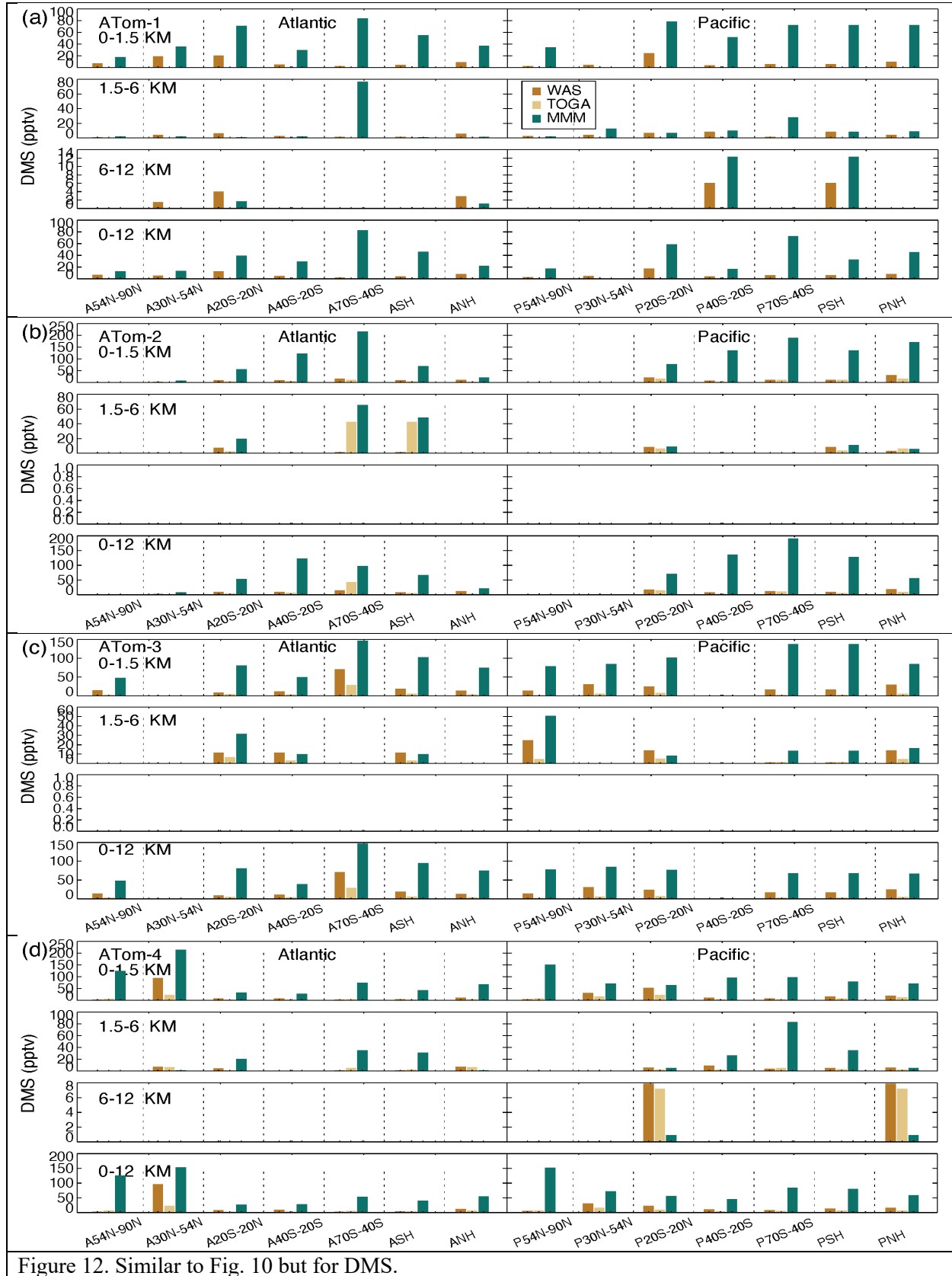

Figure 12. Similar to Fig. 10 but for DMS.


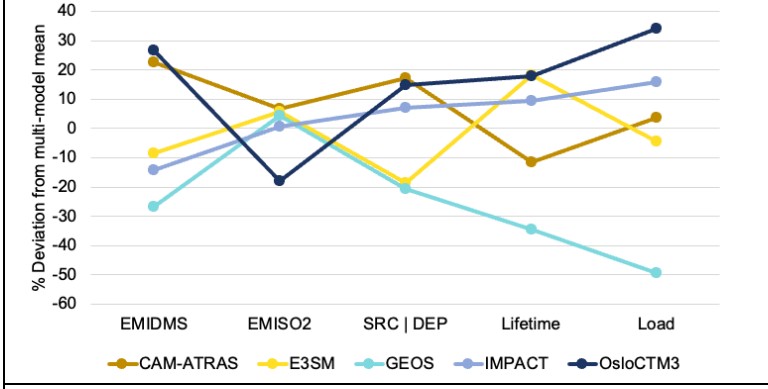

Figure 13. Deviation from multi-model mean for key budget items in sulfur study include DMS emission, SO$_2$ emission, sulfate source or total deposition, sulfate lifetime, and total sulfate atmospheric mass load.


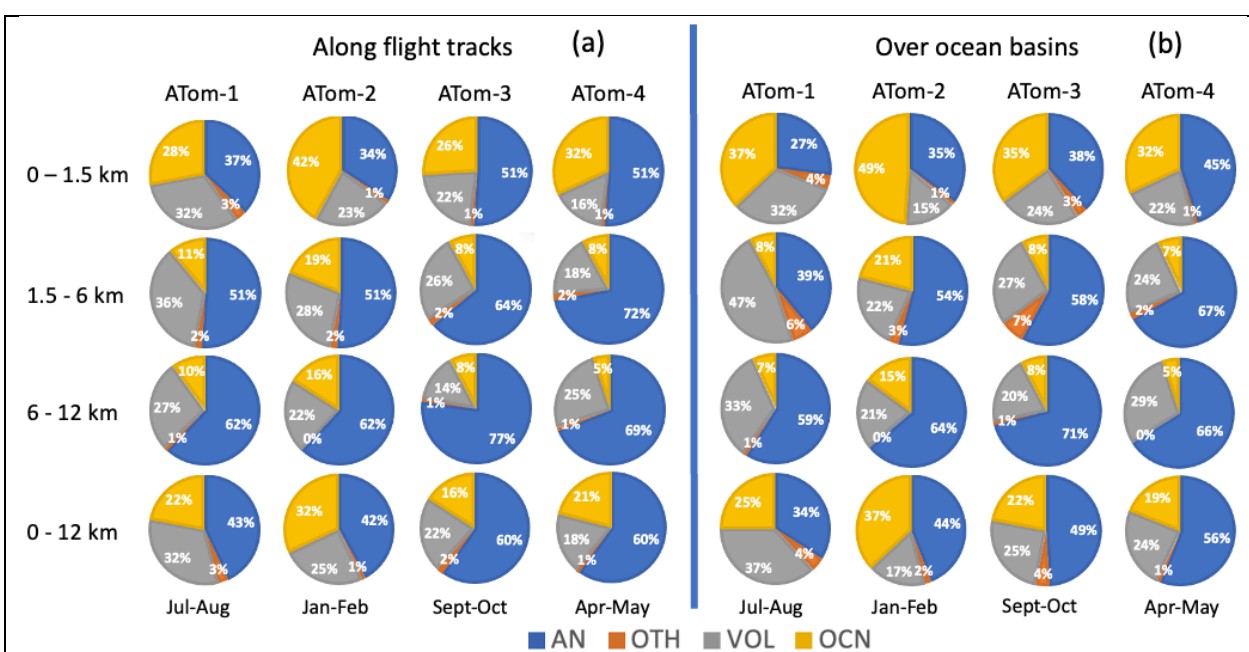

Figure 14. Source origins in percentage (%) for aerosol SO$_4$ along flight tracks (a) and for a wide oceanic area (b) based on the results from GEOS. Source origins are identified as anthropogenic (AN), volcanic (VOL), oceanic (OCN), and other sources (OTH). Ocean basins include shaded region shown in Fig. 1.
