# Peer review of "Observationally constrained analysis of sulfur cycle in the marine atmosphere with NASA ATom measurements and AeroCom model simulations"

_EGUsphere, 2023_

## Referee Comment (RC2)

**Referee Report to the article titled "Observationally constrained analysis of sulfur cycle in the marine atmosphere with NASA ATom measurements and AeroCom model simulations"**

**Summary and Scope**

The authors aim to understand sulfur cycle at remote regions in the troposphere. They use measurement data from ATom campaign and compare it with the simulated sulfur species in AeroCom chemical transport models. They discuss the possible reasons for inconsistency among models in simulating the concentrations of various sulfur species over wide ranges of conditions, and further point to possible sources of discrepancies that are avenues for further investigation. The structure of the manuscript is very logical, enabling the reader to arrive at clear inferences.

I have a few comments on the tagging method discussed in this study.

**Explicit recommendation**

I recommend **minor revisions** to certain parts of the text before accepting.

**Detailed comments**

**Main comments**

Confusion between sensitivity analysis and tagging method.

Comment 1:
In section 2.2, the authors say that they perform source attribution using sensitivity analysis method. By sensitivity analysis I would assume a perturbation approach where two simulations are performed: one with the baseline emissions and another where emissions from a given source are changed. The difference between the simulated variables in the two simulations would communicate the sensitivity of the simulated concentrations to changes in precursor emissions from a particular source.

Comment 2:
However, section 5 mentions tagging method being used for source attribution within GEOS-Chem model. Tagging involves changing the chemical mechanism within the model to write out the concentration variables along with the label of the source which they originate from.

Comment 3:
I would also like to point to Line 45 (abstract) where the authors write: "sensitivity studies by applying tagged tracers". I recommend the authors to reframe this sentence by clarifying the exact source attribution approach and model (GEOS-Chem), to avoid possible confusion among readers.

Comment 4:
Since you discuss the results from GEOS-Chem tagged simulation, please specify the tagging method in section 2.2. Cite the relevant documentation of GEOS-Chem version and other previous papers where this tagging method has been used previously if any, for attributing sulfur species to its source origins.

Comment 5:
Could you also say something about the contribution from shipping in the Pacific and Atlantic? Have you considered ship-based emissions also as anthropogenic emissions in your models? Would ship emissions which are also anthropogenic emissions be misread as anthropogenic emissions that are generally thought of as land-based emissions? Please make the necessary adjustments in their manuscripts at the locations wherever applicable (mostly sections 2.2 and section 5) and specify those changes in response to this comment.

**Minor Comments**

Comment 6:
Since you have used this tagged approach, I would also recommend that you specify the list of tags that you use. This could be either as a list in the text in section 2.2, or as a special mention in table 2.

Comment 7:
You could also add some future recommendations related to the scope of tagged simulations. With this approach we could also understand the contribution characteristics of emissions from various regions: both oceanic regions (Pacific, Atlantic, Sulfur Emission Control Areas etc.) and continental regions' (Asia, North America, South America, etc) sulfur emissions. These recommendations could be added either in section 5, where the tagged model's results are being discussed, or in the concluding section (section 6)

---

## Referee Comment (RC3)

**Review of: Observationally constrained analysis of sulfur cycle in the marine atmosphere with NASA ATom measurements and AeroCom model simulations by Bian et al.**

Bian et al. compares the spatial and temporal distribution of 4 sulfur-containing species (DMS, SO2, MSA, SO4) from the ATom airborne measurements with five global aerosol models participating in the AeroCom-ATom experiment. The study finds significant differences in the sulfur species concentrations between the models and the observations and identifies the reasons of these differences for the individual species. For DMS, the authors conclude that the emissions should be revisited, for SO2 and SO4, the authors suggest that in the models the chemical conversion of SO2 to SO4 is too rapid and that the vertical transport of SO2 is too weak. The study investigates further the model results by identifying the largest source of sulfur over ocean regions and finds that the largest contribution is dominated by anthropogenic sources followed by volcanic emissions and oceanic sources.

Given my expertise, my comments will mostly cover the modelling part and the logic of the study. The manuscript is well-written and the text well-structured. The methods and results are exposed in a logic and clear way and the whole manuscript can be easily followed and understood.

On the other hand, it is not clear what is the added value of this manuscript. The authors lack to highlight what is new in this study compared to previous literature. This is evident in the Introduction, where there is no mention of results from previous/recent studies using the ATom measurements nor using the AeroCom models. This is a major concern because the manuscript, as it is, looks like a comparison exercise with no follow-up message.
Because of this lack of novelty, I find the conclusions quite weak, and the relevance is not clearly articulated. More details are provided in the "Major comments" section below.

In light of these comments and those below, my recommendation is that the manuscript can be considered for publication in ACP once all comments below are properly addressed.

**Major Comments**

- Lack of "what's new". The manuscript does not convey the message of what is new compared to previous literature. The Introduction needs to contain references to previous literature using the Atom measurements and the AeroCom models for studies of sulfur-containing species over land and ocean. I think that one paragraph for ATom and one for AeroCom would suffice. The Abstract and the Conclusions must also state what is new in the study.

  This study can be relevant if, for instance, it is the first using the ATom airborne measurements. If this was the case, it must be emphasized considerably more in the Abstract, the Introduction (like P1L93-97 but much better/stronger) and the Conclusions (and throughout the text).
  This study can be relevant also if a comparison between AeroCom models and ATom measurements has not been done before. In this case also, the authors should emphasize the originality of such intercomparison.

- Conclusions. Based on the major comment above, the Conclusions need to be strengthened. Once the novelty of the study will be identified (again, first intercomparison?), the Conclusions must be modified accordingly. Here follow the two main conclusion points that

need to be addressed:

1) It seems to me that the authors conclude that the dominant source of atmospheric sulfur (SO2 in particular) over oceans is anthropogenic. It is well known that the largest contributions to the sulfur budget (SO2 included) is due to human activities. Hence, despite the short lifetime of the sulfur compounds, it does not seem so striking that the major source of atmospheric sulfur is anthropogenic even over oceans (e.g., Chin et al., 2000). Thus, I have the impression that this conclusion does not add anything significantly new to the current knowledge.
In order to make this conclusion more appealing, I suggest increasing its relevance by emphasizing its unicity. I couldn't find any other modeling study asserting that the sulfur over the oceans is mostly caused by human activities. This means that the authors are the first proposing that conclusion (as far as my literature research goes), and that is certainly worth mentioning. However, I suggest to carefully check the literature for modeling studies looking at the atmospheric sulfur over the oceans. If the authors find a relevant study, it would be interesting to compare their findings with yours.

2) Another conclusion consists in the description of the significant differences between the observations and the models. This is somehow discussed in the text with the attribution of the differences to emissions (DMS, e.g., P9L379-394), chemical conversion (SO2 to SO4, e.g., P10L449-451) and transport (SO2, e.g., P8L365-367) depending on the model, with a nice final discussion (P13L584588). However, these parts (especially P13L584588) can be explored further with a more detailed discussion of the models. Here's a few ideas for further discussion: 1) what are the common aspects among the models? and their differences? Can we discern something from similarities/differences 2) It is possible/feasible to perform additional sensitivity tests to assess model performances (not for this manuscript but as material for discussion)? 3) What would be the added value of satellite data in this comparison (again, for future studies): would we learn something more thanks to the denser temporal and spatial sampling compared to airborne measurements)?

3) Within the Conclusions, the authors can also dare to provide some outlook for future model developments by suggesting ways to improve the current models (e.g., update the chemical scheme, increase horizontal/vertical resolution, identify and try to improve relevant parametrization for transport).

- Tracer-tag. It would be very helpful if the authors added more detail to the methodology of the tracer-tagging concept used to obtain the source origins shown in Fig. 13. My understanding is that these sensitivity tests were conducted by the GEOS developers and made available to the authors. Even though the authors did not perform the experiments, the concept of tagged data needs to be explicitly discussed (currently it is superficially mentioned in Section 2.2) so the non-expert reader can understand the logic behind such sensitivity test.

- Figures. The figures are generally too busy – there is a lot of information that is not always entirely discussed. For example, the AMS measurements at 1s and 60s in Fig. 2 are barely discussed. In that regard, the authors state that they use 10s merged data for AMS in the text unless otherwise stated (P6L241-242). Based on that, I suggest removing the 60s and 1s lines from Figure 2 (maybe moving the 1s, 10s, 60s discussion to a separate paragraph in the

methods?). This way, the authors could merge the observations and models together in the same panels (like Fig. 3) and make Fig. 2 easier to read.

Because of the complexity of the figures, the discussion is often confusing in the sense that the authors do not specify any panel when presenting the results. For example, the analysis of Fig. 2 (P5L212-P6L264) never mentions one panel specifically. This makes it difficult to discern which flight (ATOM1-ATOM4) is considered.
In addition, the box-and-whisker panels are seldom mentioned, and they can be probably removed from there (and maybe merged and moved to Section 2 or moved to the supplement?). As a general rule, if a panel/figure is not discussed, it should not be shown.

For Figure 3, the statistics are briefly mentioned in P6L280-282.  Please provide more discussion of the statistics shown in Fig. 3 or consider moving the figures/panels to the supplement, given the large number of figures in the manuscript.

Figure 4 also contains a large number of panels – (and the box-and-whisker panels are again shortly mentioned). As my comment above, please provide more discussion or move the panels that are not discussed in the Supplement. Furthermore, the addition of panels c,f,i,l breaks the continuity with respect to the previous figures, which contain only PDFs and statistics. These panels should belong to a separate figure (and paragraph) that discusses the vertical information from the observations/model comparison (maybe merged/moved to Section 3.2?).

A final note on the box-and-whisker panels: if the authors think that the statistics should be shown, maybe it can be shown as tables depending on the sulfur species instead of panels in the figures.

For Figures 9-11, I would suggest to average together the horizontal regions that do not differ much one from another (e.g., Fig. 9a, A40S-20S with A70S-40S) and adapt the discussion accordingly.

Once the figures will be simplified, the discussion will benefit from that and there will be more room to guide the reader through the different panels of each figure.

**Minor Comments.**

- P1L1: "[…] and ecosystems.". Could you add one or two sentences on how the sulfur cycle plays the key role (e.g., pollution, acid rain)?
- P2L62-63: "wreak havoc". Could you replace this expression with something like "devastate, destroy"?
- P2L83-84: "the TUT and above are observation-sparse regions". I would rephrase this part as something like "…observations in the TUT region and above are sparse".
- P2L91: Is it possible to specify what "DC-8" means?
- P3L123: "(i.e., ~0.2-12 km) (Thompson et al., 2021)" --> (i.e., 0.2-12 km, Thompson et al., 2021).
- P4L148-150: I recommend rephrasing this sentence to something like "Two instrument were used […]: the California [….] and the NOAA [….] (Table 1)".

- P4L177: "(CMIP6) (Feng et al., 2020)" → (CMIP6, Feng et al., 2020).
- P4L178: "(…. System (GFAS))" → (… System, GFAS).
- Section 3.1: I suggest homogenizing the discussion here. As it is, Fig. 3 is less discussed than Fig. 2 and Fig. 4. The authors could enhance the discussion for Fig. 3 with the addition of a couple of paragraphs on SO2 or reduce the discussions of the Figs. 2 and 4.
- P6L252-264: The Atom flights are not discussed separately for Fig. 2 (SO4). I suggest providing some discussion about the different flights to maintain consistency with the following discussions of Figs. 3 and 4 (where the different flights are mentioned).
- P7L305: "Fig. 4c,f,i,l". This is the only place of Section 3.1 where specific panels are mentioned. Since no other panel of Figs.2, 3 or 4 is explicitly discussed, I suggest 1) either discussing explicitly the remaining panels or 2) removing the explicit mention to Fig. 4c,f,i,l and move these panels into another figure (as suggested in my major comment about Figures above).
- P7L309: I suggest removing "Apparently".
- P7L317: I suggest rephrasing "…is uniquely having a…." to "…… is unique because it has a ……"
- P8L344-345: I suggest rephrasing "Despite that improvements are needed, …" to "Despite the need for improvements, the models are generally able to capture the shape of the SO4 profile."
- P8L346-354: I find the introduction of the model Groups 1, 2 and 3 a little confusing (also, these groups are not mentioned again throughout the manuscript). I suggest keeping the model names as there are only 2 models per group and Group 2 consists only in E3SM.
- P8L358: I suggest rephrasing "…. an order of magnitude, but around …." to "…. an order of magnitude around …..".
- P8L359: The authors say that SO2 is better in IMPACT in the NH and CAM-ATRAS and OsloCTM3 in the SH, but compared to what? Please specify the reference (I suppose that would be GEOS and E3SM?).
- P8L360-361: Please be more specific with regard to how the models should improve the SO2 simulations (too large/little concentrations wrt observations?). I understand that the differences are thoroughly discussed in the following sentences, but it would be nice to have a preliminary "hint".
- P8L369-370. "All models […] ATom-1 observation". I do not quite understand what the authors mean with this sentence. Could you please explain it?
- P8L371-375. Since this part refers to a figure in the supplement (Fig. S5), I suggest reducing this discussion here.
- P9L384: "The parameterization….". Do the authors refer to the parameterization of Nightingale et al., (2000)? If yes, I suggest rewriting the beginning of the sentence with something like "That/This parameterization ….".
- P9L396: "… than the observed one," I suggest providing an example of a panel showing this steeper gradient (e.g., Fig. 7 54N-90N Atlantic). It would be nice to add panel numbers/letter to these vertical profiles to facilitate the navigation during the discussion.
- P9L399: I suggest rephrasing "tease out" with "obtain".
- P9L396-410: Very nice discussion!
- P9L414: I suggest rephrasing "These behaviors are inconsistent with…." with "These patterns do not agree with….".
- Section 3.3: Concerning the definition of the altitude ranges, I suggest keeping the same nomenclature both in the text and in the figures. Specifically, in the text the authors use

words like "free troposphere" or "boundary layer (BL)", while in the figures they show only altitude ranges in kilometers (0-1.5, 1.5-6, …. km). My suggestion is to add the name of the layer to the range in kilometer in the Figures (e.g., "0-1.5 km" becomes "0-1.5 km (BL)", etc.).

- P10L422: I suggest explaining what the Figures contain. Something like "In order to analyze [….], Figs 9-11 shows histograms of XX concentrations as a function of altitude (rows) and latitudinal band (columns)".

- P10L432-433: "The most high [….] (NH spring)." Could the authors specify the altitude range here?

- P10L438: I suggest rephrasing "more polluted" with "tends to simulate higher SO4 concentrations".

- P10L438-440: "SO4 concentrations [….] the Supplement." I suggest expanding this part by moving here the relevant parts of the Supplement.

- P10L429-440: There is no mention of the difference between Pacific and Atlantic regions. I suggest adding a couple of sentences discussing this separation.

- P10L443-444: "….. and pollution affects …." It is not clear to me what the authors mean with pollution. Could the authors explain?

- P10L444: I suggest changing "Areas where free tropospheric SO2 pollution is relatively polluted …." to "Areas where free tropospheric SO2 concentrations are relatively large …..".

- P10L442-451: there is no mention of ATom 2 in this paragraph. I suggest including one sentence discussing the most relevant pattern for ATom 2.

- P10L454: I suggest rephrasing "…. when the hemisphere is in spring" with "…. during springtime".

- P10L458-459: This sentence concerning the BL sounds a bit redundant with part of the sentence before (P10L457, discussing the free troposphere) except for the point (3). I suggest merging the discussions together to avoid repetition.

- P10L460: I suggest rephrasing "The convoluted effort can be somehow alleviated by …." with "Additional insights can be obtained by ….".

- P10L463: I suggest replacing "giving" with "because".

- P10L453-465: Also here, very nice discussion.

- P11L468: I suggest adding a sentence or two explaining the need for this Section (why it is important to look at the sulfur budget in model in light of what was shown before and what will come in Section 5).

- P11L489: I suggest rephrasing "… are pretty much the same as they should be" with "… are as expected".

- P12L524: I suggest replacing "… is at its most active" with "is the largest".

- P12L525: I suggest removing "and event".

- P12L530: I suggest removing "where and".

- P12L536 "… continental areas" I just have a question: is there any anthropogenic source of sulfur over the oceans (ships?). If yes, it would be nice to mention it.

- P12L554: "It has a clear…" What does "It" refer to? Tropospheric SO4? I suggest the authors to specify here.

- P12L556: "There are two …" This sentence needs to be introduced. Something like "Concerning volcanic sources, emissions from volcanoes are of two types."

- P12L556-P13L561: The two types of volcanic contributions need to be separated more clearly. I suggest something like "One type is the volcanic degassing [….]. The other type consists in the volcanic eruptions [….]".

- P13L560: I suggest removing "eruption".
- P13L571: I suggest adding "airborne" before "ATom".
- P13L571-573: This nice sentence is the goal and "what's new" in the manuscript. I suggest expanding this in the Introduction with references to previous studies (and a similar sentence in the Abstract). See my major comments on this aspect.
- P13L590: "over remote oceans" Again, this is the goal and should be stated more clearly in the Introduction.
- P13L599: "… this proportion is increased …" Do the authors know the amount of the increase?
- Figure 9: In the caption, I suggest adding the seasons corresponding to the different Atom flights.
- Figure 10: The contribution from biomass burning (BB) is hardly discussed in Section 5 (which is logical!). Therefore, I suggest renaming BB to something like OTHER and state that it includes biomass burning (and any other sulfur source) and that it will not be discussed in Section 5 because its contribution is negligible compared to AN, VOL and OCN.

**References.**

Chin, M., Rood, R. B., Lin, S.-J., Müller, J.-F., and Thompson, A. M. (2000), Atmospheric sulfur cycle simulated in the global model GOCART: Model description and global properties, J. Geophys. Res., 105(D20), 24671\u201324687, doi:10.1029/2000JD900384.

---

## Author Comment (AC1)

We thank the reviewers for their good summary and comments. Below is our point-to-point response.
We use PxxLxx,Fig. xx, and Table xx for revised version and OPxxLxx, OLxx, OFig.xx, OTable xx for the
original submission. Please check PxxLxx in the revised file, whose name is tagged "track".
Review 1 (R1):
**1. Relevance and novelty of this study:**
R1#1. It would be helpful if the authors could state the reasons for focusing on sulfur species in a more
straightforward way in the introduction section. While the introduction highlights the environmental
impacts of sulfur species, other pollutants not discussed in this paper also affect the environment. Why
should we focus on sulfur species? Is the bias of aerosol climate models predominantly due to sulfur?
As we mentioned in P2L74-89, the sulfur cycle plays a key role in atmospheric quality, climate,
and ecosystems. This importance is primarily due to the fact that the atmospheric sulfate
component itself contributes to radiation forcing (RF) almost as much as all other major non-
natural aerosol components, as concluded from 16 AeroCom model results (Myhre et al., 2013).
More importantly, uncertainty in sulfate simulations in current climate models is a major
contributor to biases in aerosol optical depth (AOD, Fig. 3 in Gliß et al., 2021) and radiative
forcing (RF, Fig. 7 in Myhre et al., 2013). Above discussion has been added in P2L89-94.

[Figure]

| Fig. 3 in Gliß et al., (2021): Species contributions to total AOD for each model (annual global average). | Fig. 7 in Myhre et al., (2013): Component and total RF from 16 AeroCom models. Solid lines inside the boxes show the model mean, dashed lines show the median. The box indicate one standard deviation and whiskers indicate the max and min of the distribution. |

R1#2. I would appreciate it if the authors could highlight the novelty of their work. Has any other
research analyzed the ATOM data specifically for sulfur species?
To date, there is one publication that extends the simplified DMS oxidation scheme used in
Community Atmospheric Model Chemistry version 6 (CAM6-chem) using ATom data (Fung et al.,
2022). However, a comparison of SO2, a key sulfur species in the sulfur cycle, between CAM6-
chem and ATom has not been reported. We have summarized previous studies of aerosols
(including sulfur species) related to ATom in P3L134-145.
R1#3. Is this the first study that utilizes ATOM measurements in comparison with AeroCom models?
Our work is the first study to use ATom measurements for comparison with the AeroCom
models, focusing on all sulfur species simulated in current aerosol climate models. This work extends previous efforts that utilize ATom measurements to evaluate the organic carbon (Hodzic
et al., 2020) and black carbon (Katich et al., 2018) of AeroCom models. See P3L134-137.
R1#4. If there have been studies on sulfur variability and sources over the ocean, what novel findings
does this paper present?
To our knowledge, there are no such publications investigating the variability and sources of all
major sulfur species in the ocean. Our study aims not only to reveal sulfur variability based on
multiple measurements and model simulations, but also to tease out the underlying processes
behind the variability by integrating analyzes of simulated sulfur species in aerosol climate
models. This work therefore provides direction for improvements in aerosol climate models. See
P3L141-145.
For example, our study found that all AeroCom models overestimate DMS near the sea surface
(Fig. 4). Is this overestimation due to the model DMS vertical gradient being steeper than the
ATom data (Fig. 8) or the DMS emissions being larger than they should be? A unique approach
taken in our study revealed DMS emissions as one of the causes by integrating the analysis of
DMS, SO2, and SO4 in a model-ATom comparison (P11L740-747). By reviewing the DMS
emission methods adopted by the five AeroCom models (Table 3), we further found that the
calculation of the air-sea exchange flux formula plays a key role in determining DMS emissions
(P9L589-648).
The AeroCom models also tend to underestimate SO2 and overestimate SO4 over remote
oceans. This performance prompted modelers to explore potential faster chemical conversion
of SO2 and different dry and wet removal of SO2 and SO4. One suggestion for modelers is that
using interactive oxidant calculation is insufficient to solve the problem because there is no
systematic bias in sulfate simulations (Fig. 6) between the three participating models using
online oxidants and the other two models using offline oxidants (Table 2). See P8L469-550.
R1#5. The authors should make some comparison with other studies in a new section or talk about it in
an existing section of the paper.
To date, there have been no sulfur budget reports focusing on the vast ocean. However,
previous AeroCom studies have reported global sulfate atmospheric loading and its diversity
across multiple AeroCom models using monthly and global mean column loadings. The newly
added Table 5 summarizes these studies, including their reported global and annual sulfate
multi-model mean (MMM) and diversity ($\delta$). $\delta$ is related to the standard deviation (std_dev) and
is defined as $\delta$ = std_dev / MMM *100 (%). The results of this work are lower than AeroCom-I
but higher than AeroCom-II, which may be related to the different target years involved in these
studies. One point to note is that the diversity of AeroCom-III models $\delta$ has not reduced since
AeroCom-I, which was studied nearly 20 years ago. The nearly 30% diversity combined with a
greater proportion of sulfate than other anthropogenic aerosols make accurate sulfate modeling
very important. Furthermore, these previous AeroCom studies were almost purely model
studies, with only Gliß et al., 2021 introducing AOD-related light fields from ground stations and
satellite retrievals. We have added this discussion in Sect. 4 (P12L807-823).

R1#6. Furthermore, if we improve sulfur simulation, what advantages can we expect? It would be
helpful if the authors could briefly discuss the implications of the new findings in the conclusion section.
Improved sulfur models are needed, given that the sulfur cycle plays a critical role in
atmospheric quality, climate, and ecosystems, and that its bias in current aerosol climate
models is one of the largest uncertainties in air quality and RF research (see response to R1#1).
Through a comprehensive multi-model and multi-instrument comparison of remote ocean
sulfur species in this work, shortcomings in model sulfur simulations are pointed out and
potential directions for improvement are suggested (see response to R1#4). We have added a
brief discussion of the implications of the new findings in the Conclusion section (P15L946-956),
summarizing our response to R1#1#4#5.
**2. Information Overload and Simplification:**
R1#7. Some parts of the paper contain overwhelming information that may be simplified or moved to
supplements. For instance, Section 3.1 allocates 28 lines (OL212-L239) to discuss three different
sampling intervals, which may not be key information the readers need to know. This information
(corresponding to OFig 2 (a)(d)(g)(i)) only builds up one-third content in OFigure 2, which makes the
main point of OFigure 2 very hard to catch.
The discussion of the three-time resolution AMS data (OL212-239) and their visualization in Fig.
2 have been moved to Supplementary Information. We have now made Fig. 2 in a manner
consistent with Figs. 3-4.
R1#8. Another example starts from OL317 where the authors spent time explaining how the flag '-888' is
replaced by '0' to represent the low values, which, although crucial for validating results, may not be
necessary for most readers.
OL317-329 have also been moved to Supplementary Information.
R1#9. Additionally, the division of 5 models into 3 groups from OL340-L354 seems superfluous and is
never referenced throughout the paper. I would recommend a description without grouping the models.
These sentences are changed to "Specifically, CAM-ATRAS and GEOS have good SO4 vertical gradients
over the tropical and NH oceans, but their SO4 values are too low compared to measurements over the
Southern Hemisphere (SH) free troposphere. The SO4 of IMPACT and OsloCTM3 decreases too slowly
with altitude, as shown by their overestimated SO4 values at high altitudes globally. E3SM performed
SO4 simulations among other models. However, the performance of these models' SO4 vertical profiles
cannot simply be explained by the way the oxidant is applied, because among the five models, CAM-
ATRAS, IMPACT, and OsloCTM3 used interactive oxidant calculations, while E3SM and GEOS used
archived oxidant data (Table 2). The complexity of the chemistry deserves more attention. Of the five
models, OsloCTM3 and GEOS participated in the multi-model OH assessment (Nicely et al., 2000) and
OsloCTM3 had a shorter methane lifetime (relative to OH) than GEOS." See P8L469-550.
**3. Layout and Readability of Figures:**
R1#10. Due to the huge load of information that is shown, optimizing the layout of figures is crucial to
enhance readability. For example, I would recommend relocating the legends in OFigures 5-8 and
putting this information on the top/bottom or right side of the charts.
The legends in Figs. 6-9 (OFigs. 5-8) are now at the top of the charts.

R1#11. Since you 'use 10-s merged data where observations above DL throughout the main text unless
otherwise stated' (OL241), could you just show the results of 10-s data only on OFigure 2 and move the
other to the supplement?
Done. Please also see response to R1#7.
R1#12. Additionally, as Atom-1/2/3/4 are not following the order of the four seasons, I recommend
adding notes on the seasons at a proper place in OFigures 9,10,11, and 13 to guide the readers when
reading through the section about seasonal changes in the paper.
Done.

**4. Conclusion:**

R1#13. The conclusion section is mostly a summary of the content. As mentioned earlier, the implication
of the new findings can be stated in this section.
We have added this paragraph in the Conclusion section (P15L946-956):
"Even after two decades of development, the diversity of sulfate simulations from AeroCom-I to
AeroCom-III has not decreased. However, accurate sulfate simulation in current climate models is crucial
to reduce radiative forcing biases. Several potential directions for improving sulfur simulations are
suggested above. More importantly, apart from the shortcomings of individual models, all modelers
should focus on the calculation of the air-sea exchange flux formula, as it plays a key role in determining
DMS emissions. Modelers also need to study DMS and SO2 vertical transport as well as SO4 wet
deposition during long-distance transport, as model biases are greatest at high altitudes. One suggestion
to modelers is that the use of online oxidant fields is insufficient to explain the model sulfate bias, as
there was no systematic bias in the sulfate simulations between the models using interactive oxidants
and the models using archival oxidants in this study. The complexity of chemistry deserves more
attention."
Please also see response to R1#4.

**Technical comments:**

1.  Please standardize the color and font of the indices of panels in OFigure 2.
Indices of panels in Fig. 2 have been removed based on the suggestion of reviewer 3 (R3P7L305).
2.  Please refine OFigure 12 to maintain the consistent style of other figures.
Done.
3.  In the caption of OFigure 9, AMS should be orange instead of 'red'.
Done.
4.  Please replace the vertical bar in OFigure 13 with a straight line as the shape and color is
misleading.
Done.
5.  In OL467, please add a period after '4'.
Done.

**Reviewer 2 (R2):**

R2#1. In section 2.2, the authors say that they perform source attribution using sensitivity analysis
method. By sensitivity analysis I would assume a perturbation approach where two simulations are
performed: one with the baseline emissions and another where emissions from a given source are
changed. The difference between the simulated variables in the two simulations would communicate the sensitivity of the simulated concentrations to changes in precursor emissions from a particular
source.
Yes, we need two simulated fields, one linked to baseline emission and the other linked to a given
source, to diagnose source attribution. We have two methods to calculate the two fields in model
simulation.
     1. Run model twice for field (F). The first run F is linked to the baseline emission (F(base-emi)) and
        the second run F is linked to a given source (F(tag-emi)).
     2. Run model once for fields F and Ftag (a newly added tracer) simultaneously. F is linked to the
        baseline emission and Ftag is linked to a given source.
The advantage of using the second method is obvious, simply for computational efficiency, especially
when multiple tagged tracers are required. See newly added Sect. 2.3 "Tag-tracer study in GEOS".

R2#2. However, section 5 mentions tagging method being used for source attribution within GEOS-Chem
model. Tagging involves changing the chemical mechanism within the model to write out the
concentration variables along with the label of the source which they originate from.
Each specific aerosol component in GEOS GOCART is simulated independently from the others, and the
contribution of each emission type to the total aerosol mass is also not interfered by that of other
emission types. Thus, additional aerosol tracers can easily be "tagged" according to emission source
types. The GEOS GOCART module includes a tagged aerosol mechanism that allows GOCART to calculate
and transfer multiple sets of aerosol fields (e.g., A, Atag, etc) in a single job submission. See newly added
Sect. 2.3 "Tag-tracer study in GEOS".

R2#3. I would also like to point to Line 45 (abstract) where the authors write: "sensitivity studies by
applying tagged tracers". I recommend the authors to reframe this sentence by clarifying the exact
source attribution approach and model (GEOS-Chem), to avoid possible confusion among readers.
Changed the sentence to "We identify their origins from anthropogenic versus natural sources with
sensitivity studies by applying tagged tracers in GEOS model linking to emission types of anthropogenic,
biomass burning, volcanic, and oceanic emissions." See P1L44-47.

R2#4. Since you discuss the results from GEOS-Chem tagged simulation, please specify the tagging
method in section 2.2. Cite the relevant documentation of GEOS-Chem version and other previous
papers where this tagging method has been used previously if any, for attributing sulfur species to its
source origins.
The description of the tagging method has been added in Sect. 2.3.
Tracer-tag technical in GEOS has been widely used in aerosol and gas studies (Bian et al., 2021; Nielsen et
al., 2017; Strode et al., 2018) and in supporting various aircraft field campaigns such as ARCTAS, KORUS-
AQ, ATom, CAMP2Ex, ACCLIP, and more. Such techniques are also adopted in other models such as
GEOS-Chem model (Fisher et al., 2017; Ikeda et al., 2017) and CESM (Butler et al., 2018).

R2#5: Could you also say something about the contribution from shipping in the Pacific and Atlantic?
Have you considered ship-based emissions also as anthropogenic emissions in your models? Would ship
emissions which are also anthropogenic emissions be misread as anthropogenic emissions that are
generally thought of as land-based emissions? Please make the necessary adjustments in their
manuscripts at the locations wherever applicable (mostly sections 2.2 and section 5) and specify those
changes in response to this comment.
Shipping emissions are concentrated between the tropics and mid-latitudes of the Northern Hemisphere
over the Pacific and Atlantic Oceans. In our SO4 source analysis (Sect. 5), we discussed the contribution
of various emission types (anthropogenic, biomass burning, volcanic, and marine) to atmospheric SO4

over remote oceans. Here shipping emission was considered as anthropogenic emissions. We did not
further differentiate the location of anthropogenic emissions in this study. We specifically pointed this
out in Sect. 2.2 (P5L231-233) and modified accordingly the Abstract P1L44-46, Introduction P3L130-132,
main text P13L852-853, and Conclusion P15L929-930.

R2#6. Since you have used this tagged approach, I would also recommend that you specify the list of tags
that you use. This could be either as a list in the text in section 2.2.
Four tags we used in GEOS for this study relate to emission types of anthropogenic, biomass burning,
volcanic, and oceanic emissions. This list is now given in Abstract P1L46-47 and Sect. 2.3.

R2#7: You could also add some future recommendations related to the scope of tagged simulations.
With this approach we could also understand the contribution characteristics of emissions from various
regions: both oceanic regions (Pacific, Atlantic, Sulfur Emission Control Areas etc.) and continental
regions' (Asia, North America, South America, etc) sulfur emissions. These recommendations could be
added either in section 5, where the tagged model's results are being discussed or in the concluding
section (section 6).
Tags can be linked to various emission types (anthropogenic, biomass burning, etc.) and/or various
emission regions (Pacific, Atlantic, Asia, North America, etc.), given that the contribution of each
emission type/region to the total aerosol mass is also not interfered by that of other emission
types/regions. The tagged simulation is a powerful tool that is widely used not only in aerosol
components (Bian et al., 2021; Ikeda et al., 2017), but also in tracer gases such as CO (Fisher et al., 2017;
Strode et al., 2018; Nielsen et al., 2017), CO2 (Lin et al., 2020), and O3 (Butler et al., 2018). This
technique is particularly useful when supporting aircraft field campaigns (ARCTAS, KORUS-AQ, ATom,
CAMP2Ex, ACCLIP, and etc) for flight planning due to its computational efficiency and its characteristic of
tracking the origin of plumes linking to emission type and location. We have added this discussion in
Sect. 2.3.

R3#1.The authors lack to highlight what is new in this study compared to previous literature. This is
evident in the Introduction, where there is no mention of results from previous/recent studies using the
ATom measurements nor using the AeroCom models. This is a major concern because the manuscript, as
it is, looks like a comparison exercise with no follow-up message.
Thanks for this comment. In the introduction, we have added a paragraph (P3L134-141) summarizing
relevant previous/recent studies using the ATom measurements and the AeroCom models.

R3#2. Lack of "what's new". The manuscript does not convey the message of what is new compared to
previous literature. The Introduction needs to contain references to previous literature using the Atom
measurements and the AeroCom models for studies of sulfur-containing species over land and ocean. I
think that one paragraph for ATom and one for AeroCom would suffice. The Abstract and the Conclusions
must also state what is new in the study.
Yes. We have added a paragraph reviewing the literature and noting "what's new" in P3L134-145 (see
also reply to R3#1).

Our abstract now includes this sentence (P1L47-49): "Our work presents the first assessment of
AeroCom sulfur study using ATom measurements, providing directions for improving sulfate simulations,
which remain the largest uncertainty in radiative forcing estimates in aerosol climate models."

The Conclusions now include this paragraph (P15L946-956): "Even after two decades of development,
the diversity of sulfate simulations from AeroCom-I to AeroCom-III has not decreased. However, accurate
sulfate simulation in current climate models is crucial to reduce radiative forcing biases. Several potential
directions for improving sulfur simulations are suggested above. More importantly, apart from the
shortcomings of individual models, all modelers should focus on the calculation of the air-sea exchange
flux formula, as it plays a key role in determining DMS emissions. Modelers also need to study DMS and
SO2 vertical transport as well as SO4 wet deposition during long-distance transport, as model biases are
greatest at high altitudes. One suggestion to modelers is that the use of online oxidant fields is
insufficient to explain the model sulfate bias, as there was no systematic bias in the sulfate simulations
between the models using interactive oxidants and the models using archival oxidants in this study. The
complexity of chemistry deserves more attention."

This study can be relevant if, for instance, it is the first using the ATom airborne measurements. If this
was the case, it must be emphasized considerably more in the Abstract, the Introduction (like OP1L93-97
but much better/stronger) and the Conclusions (and throughout the text).
No, ATom measurements have been used in various studies. The ATom aerosol studies include NPF in the
tropics (Williamson et al., 2019), fine aerosol lifetimes (Gao et al., 2022), and various aerosol
components such as sea salt (Bian et al., 2019), OA ( Hodzic et al., 2020), smoke (Schill et al., 2020),
mineral dust (Froyd et al., 2022), and DMS chemistry (Fung et al., 2022). See P3L134-140.

This study can be relevant also if a comparison between AeroCom models and ATom measurements has
not been done before. In this case also, the authors should emphasize the originality of such
intercomparison.
Our work is the first study to use ATom measurements for comparison with the AeroCom models,
focusing on all sulfur species simulated in current aerosol climate models. This work extends a previous
effort that utilizes ATom measurements for comparison with AeroCom models for organic aerosol
(Hodzic et al., 2020). See new paragraph in P3L134-145.

R3#3. Conclusions. Based on the major comment above, the Conclusions need to be strengthened. Once
the novelty of the study will be identified (again, first intercomparison?), the Conclusions must be
modified accordingly. Here follow the two main conclusion points that need to be addressed:

1)  It seems to me that the authors conclude that the dominant source of atmospheric sulfur (SO2
in particular) over oceans is anthropogenic. It is well known that the largest contributions to the
sulfur budget (SO2 included) is due to human activities. Hence, despite the short lifetime of the
sulfur compounds, it does not seem so striking that the major source of atmospheric sulfur is
anthropogenic even over oceans (e.g., Chin et al., 2000). Thus, I have the impression that this
conclusion does not add anything significantly new to the current knowledge. In order to make
this conclusion more appealing, I suggest increasing its relevance by emphasizing its unicity. I
couldn't find any other modeling study asserting that the sulfur over the oceans is mostly
caused by human activities. This means that the authors are the first proposing that conclusion
(as far as my literature research goes), and that is certainly worth mentioning. However, I
suggest to carefully check the literature for modeling studies looking at the atmospheric sulfur
over the oceans. If the authors find a relevant study, it would be interesting to compare their
findings with yours.
The authors also could not find other modeling study asserting that the sulfur over the oceans is
mostly caused by human activities. We changed the conclusion sentence of "On the other hand, our study shows that anthropogenic emissions remain a major source of sulfate aerosols
generated over remote oceans ..." to "On the other hand, our study is the first asserting that
anthropogenic emissions remain a major source of sulfate aerosols generated over remote
oceans ..." (P15L941-944).

2)  Another conclusion consists in the description of the significant differences between the
observations and the models. This is somehow discussed in the text with the attribution of the
differences to emissions (DMS, e.g., OP9L379-394), chemical conversion (SO2 to SO4, e.g.,
OP10L449-451) and transport (SO2, e.g., OP8L365-367) depending on the model, with a nice
final discussion (OP13L584588). However, these parts (especially OP13L584588) can be explored
further with a more detailed discussion of the models. Here's a few ideas for further discussion:
1) what are the common aspects among the models? and their differences? Can we discern
something from similarities/differences 2) It is possible/feasible to perform additional sensitivity
tests to assess model performances (not for this manuscript but as material for discussion)? 3)
What would be the added value of satellite data in this comparison (again, for future studies):
would we learn something more thanks to the denser temporal and spatial sampling compared
to airborne measurements)?
Thanks for these excellent suggestions.
Regarding recommendation 1: We examined the DMS emission derivation methods used in five
AeroCom models (see the newly added Table 3 and the corresponding discussion in P9L573-581) and
found that the parameterization of air-sea exchanges in DMS flux calculations is very important P9L589-
648). We also recognized that using interactive oxidant calculation is insufficient to account for the
model sulfate bias (see P8L473-550).

Regarding recommendation 2: Yes. Several sensitivity tests have been suggested in Conclusion P14L910-
915.

Regarding recommendation 3: In-situ aircraft measurements provide intensive and extensive fields that
can be used to improve model simulations. However, aircraft measurements are limited by space and
time. On the other hand, satellite data coupled with global long-term measurements can provide an
independent assessment of model improvements. In this way, we can use the model as a bridge
between aircraft and satellite measurements. Denser temporal and spatial sampling of aircraft
measurements can provide tracer vertical profiles, which are critical for modeling tracer global
distribution and assisting satellite AOD/AI retrievals.

Additionally, aircraft measurements can provide detailed component measurements that cannot
currently be retrieved from satellite measurements. More intensive temporal aircraft measurements can
also provide tracer diurnal cycles, which are important for air quality studies.

3) Within the Conclusions, the authors can also dare to provide some outlook for future model
developments by suggesting ways to improve the current models (e.g., update the chemical scheme,
increase horizontal/vertical resolution, identify and try to improve relevant parametrization for
transport).
Several potential directions for improving sulfur simulations were suggested in Conclusion P14L910-915.
We also added the following sentences (P15L949-956) to the Conclusion. "More importantly, apart from
the shortcomings of individual models, all modelers should focus on the calculation of the air-sea
exchange flux formula, as it plays a key role in determining DMS emissions. Modelers also need to study

DMS and SO2 vertical transport as well as SO4 wet deposition during long-distance transport, as model
biases are greatest at high altitudes. One suggestion to modelers is that the use of online oxidant fields is
insufficient to explain the model sulfate bias, as there was no systematic bias in the sulfate simulations
between the models using interactive oxidants and the models using archival oxidants in this study. The
complexity of chemistry deserves more attention."
R3#4. Tracer-tag. It would be very helpful if the authors added more detail to the methodology of the
tracer-tagging concept used to obtain the source origins shown in OFig. 13. My understanding is that
these sensitivity tests were conducted by the GEOS developers and made available to the authors. Even
though the authors did not perform the experiments, the concept of tagged data needs to be explicitly
discussed (currently it is superficially mentioned in Section 2.2) so the non-expert reader can understand
the logic behind such sensitivity test.
The tracer-tag concept is now described in Sect 2.3 "Tag-tracer study in GEOS". Please also see response
to R2#1 - #4 and R2#6 - #7.
R3#5.
Figures. The figures are generally too busy – there is a lot of information that is not always entirely
discussed. For example, the AMS measurements at 1s and 60s in OFig. 2 are barely discussed. In that
regard, the authors state that they use 10s merged data for AMS in the text unless otherwise stated
(OP6L241-242). Based on that, I suggest removing the 60s and 1s lines from OFigure 2 (maybe moving
the 1s, 10s, 60s discussion to a separate paragraph in the methods?). This way, the authors could merge
the observations and models together in the same panels (like OFig. 3) and make OFig. 2 easier to read.
The information of 60-s and 1-s in OFig. 2 and corresponding discussion has been moved to the
Supplementary Information.
Because of the complexity of the figures, the discussion is often confusing in the sense that the authors
do not specify any panel when presenting the results. For example, the analysis of Fig. 2 (OP5L212-
P6L264) never mentions one panel specifically. This makes it difficult to discern which flight (ATOM1-
ATOM4) is considered.
Since the discussion follows the number of ATom deployments given in the panels, we removed the
panel indices from Fig. 2.
In addition, the box-and-whisker panels are seldom mentioned, and they can be probably removed from
there (and maybe merged and moved to Section 2 or moved to the supplement?). As a general rule, if a
panel/figure is not discussed, it should not be shown.
Discussion L350-355 is based on the information shown in the box-and-whisker panels.
For OFigure 3, the statistics are briefly mentioned in OP6L280-282. Please provide more discussion of
the statistics shown in OFig. 3 or consider moving the figures/panels to the supplement, given the large
number of figures in the manuscript.
The discussion has been modified as "Statistics indicate lower model SO2 medians than observed (box-
and-whisker in Fig. 3), especially during ATom-1. However, the model means are comparable or even
higher than those observed, indicating that the models simulate unobserved episode events.
Consequently, the simulated mean/median ratio is higher than the observed value. Across the four
ATom deployments, ATom-4 has much better model observation consistency." See P7L385-389.

OFigure 4 also contains a large number of panels – (and the box-and-whisker panels are again shortly
mentioned). As my comment above, please provide more discussion or move the panels that are not
discussed in the Supplement. Furthermore, the addition of panels c,f,i,l breaks the continuity with
respect to the previous figures, which contain only PDFs and statistics. These panels should belong to a
separate figure (and paragraph) that discusses the vertical information from the observations/model
comparison (maybe merged/moved to Section 3.2?).
The discussion of P8L435-441 is based on the statistics shown by box-and-whisker panels. We pointed
out this in the revised version.
The OFig. 4 has been separated into two figures: one contains PDFs and statistics (Fig. 4) and the other
contains original panels c,f,I,l (Fig. 5).
A final note on the box-and-whisker panels: if the authors think that the statistics should be shown,
maybe it can be shown as tables depending on the sulfur species instead of panels in the figures.
To avoid the graph being too busy, we removed the values of median and mean from the graph.
However, we do think that summarizing the various statistics in charts can help readers navigate the
data more easily.
For OFigures 9-11, I would suggest to average together the horizontal regions that do not differ much
one from another (e.g., Fig. 9a, A40S-20S with A70S-40S) and adapt the discussion accordingly.
The sulfur species can differ much between A40S-20S and A70S-40S depending on the season. For
example, the Atlantic free tropospheric SO2 over the A40S-20S in ATom-4 is much higher than over the
A70S-40S (Fig. 10d). Such performance is also visible in SO4 (Fig. 9d) but to a much smaller extent.
However, this behavior is not shown across the entire vertical column of DMS (Fig. 11d), indicating that
a marine source is not the cause. Considering that A40S-20S is located in the downwind area of South
America (SA), SA pollution showed an impact on SO2 in the marine atmosphere of A40S-20S. See
discussion in P11L719-723.
**R3 Minor Comments.**
• OP1L1: "[…] and ecosystems.". Could you add one or two sentences on how the sulfur cycle plays
the key role (e.g., pollution, acid rain)?
Is this typo for P1L34? Changed the sentence to "The sulfur cycle plays a key role in atmospheric air
quality, climate, and ecosystems, such as pollution, radiative forcing, new particle formation, and
acid rain."
• OP2L62-63: "wreak havoc". Could you replace this expression with something like "devastate,
destroy"?
Done.
• OP2L83-84: "the TUT and above are observation-sparse regions". I would rephrase this part as
something like "…observations in the TUT region and above are sparse".
Done.
• OP2L91: Is it possible to specify what "DC-8" means?
DC-8 means Douglas DC-8 jetliner. See change in P3L116.
• OP3L123: "(i.e., ~0.2-12 km) (Thompson et al., 2021)" --> (i.e., 0.2-12 km, Thompson et al., 2021).
Done.
• OP4L148-150: I recommend rephrasing this sentence to something like "Two instrument were used
[…]: the California [….] and the NOAA [….] (Table 1)".

447  Done.

• OP4L177: "(CMIP6) (Feng et al., 2020)" →(CMIP6, Feng et al., 2020).

449  Changed the sentence to "The suggested emissions are the Coupled Model Intercomparison Project
450  Phase 6 Community Emissions Data System (CEDS, Hoesly et al., 2018) ….". See P5L224-227.

• OP4L178: "(…. System (GFAS))" →(… System, GFAS).

452  Done.

• Section 3.1: I suggest homogenizing the discussion here. As it is, OFig. 3 is less discussed than OFig. 2
454  and OFig. 4. The authors could enhance the discussion for OFig. 3 with the addition of a couple of
455  paragraphs on SO2 or reduce the discussions of the OFigs. 2 and 4.

456  The discussion of Figs. 2 and 4 has been reduced in Sect. 3.1.

• OP6L252-264: The Atom flights are not discussed separately for OFig. 2 (SO4). I suggest providing
458  some discussion about the different flights to maintain consistency with the following discussions of
459  OFigs. 3 and 4 (where the different flights are mentioned).

460  We added discussion of different flights in P7L363-366. We also reframed the paragraph by
461  discussing two measures first. This style is consistent with the style used in SO2 and DMS (i.e., Figs.
462  3-4) discussions.

• OP7L305: "OFig. 4c,f,i,l". This is the only place of Section 3.1 where specific panels are mentioned.
465  Since no other panel of OFigs.2, 3 or 4 is explicitly discussed, I suggest 1) either discussing explicitly
466  the remaining panels or 2) removing the explicit mention to OFig. 4c,f,i,l and move these panels into
467  another figure (as suggested in my major comment about OFigures above).

468  Following the reviewer' suggestion, we have removed the indices of the panels in Figs. 2-4 and
469  discussed the figures following ATom-# given in the figures. We also extracted OFig. 4c,f,l,l and put
470  them in Fig. 5 to maintain continuity among Figs. 2-4.

• OP7L309: I suggest removing "Apparently".

472  Done.

• OP7L317: I suggest rephrasing "…is uniquely having a…." to "…… is unique because it has a ……"
474  Done.

• OP8L344-345: I suggest rephrasing "Despite that improvements are needed, …" to "Despite the
476  need for improvements, the models are generally able to capture the shape of the SO4 profile."

477  Done.

• OP8L346-354: I find the introduction of the model Groups 1, 2 and 3 a little confusing (also, these
479  groups are not mentioned again throughout the manuscript). I suggest keeping the model names as
480  there are only 2 models per group and Group 2 consists only in E3SM.

481  Removed the group categories, see P8L469-473. Also see R1#9 and R3#3.

• OP8L358: I suggest rephrasing "…. an order of magnitude, but around …." to "…. an order of
483  magnitude around ….".

484  Done.

• OP8L359: The authors say that SO2 is better in IMPACT in the NH and CAM-ATRAS and OsloCTM3 in
486  the SH, but compared to what? Please specify the reference (I suppose that would be GEOS and
487  E3SM?).

488  Compared to other AeroCom models not mentioned in the sentence, see P9L553-555.

• OP8L360-361: Please be more specific with regard to how the models should improve the SO2
490  simulations (too large/little concentrations wrt observations?). I understand that the differences are
491  thoroughly discussed in the following sentences, but it would be nice to have a preliminary "hint".

Yes. The sentence is changed to "The tropical Pacific appears to be an interesting region, with all
models except GEOS failing to capture observed local SO2 sources." See P9L556-557.

• OP8L369-370. "All models [....] ATom-1 observation". I do not quite understand what the authors
mean with this sentence. Could you please explain it?

The sentence is changed to "All other models show lower SO2 at the surface than in the lower free
troposphere, which is inconsistent with the observed profiles." See PL9565-566.

• OP8L371-375. Since this part refers to a figure in the supplement (OFig. S5), I suggest reducing this
discussion here.

The following sentence (OP8L372-375) was moved to the Supplementary Information: "Modeled
SO2 volume mixing ratios are generally lower compared to the CIMS observations for most altitudes
and latitudinal bins in ATom-1 to -3, which may be partially owing to the CIMS measurement issue
discussed in Sect. 3.1.".

• OP9L384: "The parameterization....". Do the authors refer to the parameterization of Nightingale et
al., (2000)? If yes, I suggest rewriting the beginning of the sentence with something like "That/This
parameterization ....".

The paragraph has been rewritten to response R3#3, so this comment is no long relevant in the
revised version.

• OP9L396: "… than the observed one," I suggest providing an example of a panel showing this
steeper gradient (e.g., OFig. 7 54N-90N Atlantic). It would be nice to add panel numbers/letter to
these vertical profiles to facilitate the navigation during the discussion.

We added "(e.g., Fig. 8 A54N-90N)" to P10L650-651. However, if we add panel numbers to these
vertical profiles, we will end up mentioning only one number in the text, which goes against the
reviewer's suggestion (see P7L305 above).

• OP9L399: I suggest rephrasing "tease out" with "obtain".

Done.

• OP9L396-410: Very nice discussion!

Thanks.

• OP9L414: I suggest rephrasing "These behaviors are inconsistent with...." with "These patterns do
not agree with....".

Done.

• Section 3.3: Concerning the definition of the altitude ranges, I suggest keeping the same
nomenclature both in the text and in the figures. Specifically, in the text the authors use words like
"free troposphere" or "boundary layer (BL)", while in the figures they show only altitude ranges in
kilometers (0-1.5, 1.5-6, …. km). My suggestion is to add the name of the layer to the range in
kilometer in the Figures (e.g., "0-1.5 km" becomes "0-1.5 km (BL)", etc.).
Done.

• OP10L422: I suggest explaining what the Figures contain. Something like "In order to analyze [....],
Figs 9-11 shows histograms of XX concentrations as a function of altitude (rows) and latitudinal band
(columns)".

The sentence has been changed to "In order to analyze model performance on a regional and
seasonal basis, Figs. 10-12  show histograms of SO4, SO2, and DMS concentrations as a function of
altitude (rows) and latitudinal band (columns)." See P10L676-678.

• OP10L432-433: "The most high [....] (NH spring)." Could the authors specify the altitude range here?

The altitude range for free-troposphere refers to 1.5 – 12 km, see change in P10L684-685.

• OP10L438: I suggest rephrasing "more polluted" with "tends to simulate higher SO4
concentrations".

Done.

• OP10L438-440: "SO4 concentrations [....] the Supplement." I suggest expanding this part by moving
here the relevant parts of the Supplement.

The relevant discussion of the Supplement (see below) has been moved to the main text.

P11L711-715: "For example, in summer and winter, the CAM-ATRAS model gave the highest
estimates of atmospheric SO4 in the oceanic BL, but the IMPACT and OsloCTM3 models gave the
highest estimates of atmospheric SO4 in the free troposphere (Fig. S9). All models except the GEOS
model generally overestimate SO4 in the atmosphere."

P11L727-732: "For example, the E3SM model gives significantly higher SO2 compared with the
measurements and other models in BL (Fig. S10). Unlike the case of SO4, all models tend to
underestimate SO2 in the free troposphere, with some exceptions such as the GEOS model in the
North Pacific mid-high-latitude winter (ATom-2) and the CAM-ATRAS and IMAPCT models in the
South Atlantic mid-latitude autumn (ATom-4)."

P11L745-747: "The overestimation of the DMS multi-model median in Fig. 12 is clearly attributable
to the contribution of all models shown in Fig. S11, with the models CAM-ATRAS and OsloCTM3
being more prominent."

• OP10L429-440: There is no mention of the difference between Pacific and Atlantic regions. I suggest
adding a couple of sentences discussing this separation.

We added following sentences in P10L692-709: "Compared to observations, model tends to
simulate higher SO4 concentrations in the free tropospheric atmosphere. Both observations and
simulations show that the SO4 in the Pacific is higher than that in the Atlantic during the NH high-
latitude autumn (ATom-3) and the NH mid-latitude spring (ATom-4). The differences between
observations and simulations are generally larger in the Atlantic than in the Pacific, particularly in
the SH."

• OP10L443-444: "….. and pollution affects …." It is not clear to me what the authors mean with
pollution. Could the authors explain?

Changed "pollution affects" to "this high SO2 region extends to". See P11L718-719.

• OP10L444: I suggest changing "Areas where free tropospheric SO2 pollution is relatively polluted …."
to "Areas where free tropospheric SO2 concentrations are relatively large …..".

Done.

• OP10L442-451: there is no mention of ATom 2 in this paragraph. I suggest including one sentence
discussing the most relevant pattern for ATom 2.

The discussion for ATom-2 has been added in P11L720-723. "For instance, free troposphere appears
to be more polluted than other regions in the NH Pacific during ATom-2 and in the SH mid-latitude
Atlantic during ATom-4, but not in the BL, implying a potential source of horizontal transport."

• OP10L454: I suggest rephrasing "…. when the hemisphere is in spring" with "…. during springtime".

Done.

• OP10L458-459: This sentence concerning the BL sounds a bit redundant with part of the sentence
before (P10L457, discussing the free troposphere) except for the point (3). I suggest merging the
discussions together to avoid repetition.

Removed this sentence (OP10L457-458) to avoid repetition: "… suggesting a potential slower
vertical transport or faster DMS chemical loss in models."

• OP10L460: I suggest rephrasing "The convoluted effort can be somehow alleviated by …." with
"Additional insights can be obtained by ….".
Done.
• OP10L463: I suggest replacing "giving" with "because".
Done.
• OP10L453-465: Also here, very nice discussion.
Thanks.
• OP11L468: I suggest adding a sentence or two explaining the need for this Section (why it is
important to look at the sulfur budget in model in light of what was shown before and what will
come in Section 5).
We added following two sentences there: "Budget analysis is a simple and basic method that has
been widely used to document the underlying performance of a model. This analysis allows us to
evaluate the AeroCom-III sulfur simulations against previous AeroCom-I and -II studies and reserves
a record for future model evaluations." See P11L750-768.
• OP11L489: I suggest rephrasing "… are pretty much the same as they should be" with "… are as
expected".
Changed the sentence to "source and deposition are pretty much the same as expected".
• OP12L524: I suggest replacing "… is at its most active" with "is the largest".
Done.
• OP12L525: I suggest removing "and event".
Done.
• OP12L530: I suggest removing "where and".
Done.
• OP12L536 "… continental areas" I just have a question: is there any anthropogenic source of sulfur
over the oceans (ships?). If yes, it would be nice to mention it.
Done. See P13L852-853.
• OP12L554: "It has a clear…" What does "It" refer to? Tropospheric SO4? I suggest the authors to
specify here.
Changed "It has a clear …" to "There is a clear …".
• OP12L556: "There are two …" This sentence needs to be introduced. Something like "Concerning
volcanic sources, emissions from volcanoes are of two types."
Done.
• OP12L556-P13L561: The two types of volcanic contributions need to be separated more clearly. I
suggest something like "One type is the volcanic degassing [….]. The other type consists in the
volcanic eruptions [….]".
Done. See P14L882-886.
• OP13L560: I suggest removing "eruption".
Done.
• OP13L571: I suggest adding "airborne" before "ATom".
Done.
• OP13L571-573: This nice sentence is the goal and "what's new" in the manuscript. I suggest
expanding this in the Introduction with references to previous studies (and a similar sentence in the
Abstract). See my major comments on this aspect.

Yes. We have added a paragraph reviewing previous relevant research in the Introduction section
(P3L134-145). We have also added Table 5 and corresponding discussion (P12L807-823) comparing
the diversity of global sulfate atmospheric loads from multiple AeroCom models since Aerocom-I.
See also responses to R1#5 and R3#2.

• OP13L590: "over remote oceans" Again, this is the goal and should be stated more clearly in the
Introduction.

These key words have been mentioned already in Abstract (P1L40), the Introduction (P2L105-106,
L121, L128-129), and model experiment design (P5L250).

• OP13L599: "… this proportion is increased …" Do the authors know the amount of the increase?

This SO4 increased by explosive eruptions varies in location and timing.

• OFigure 9: In the caption, I suggest adding the seasons corresponding to the different Atom flights.

Done.

• OFigure 10: The contribution from biomass burning (BB) is hardly discussed in Section 5 (which is
logical!). Therefore, I suggest renaming BB to something like OTHER and state that it includes
biomass burning (and any other sulfur source) and that it will not be discussed in Section 5 because
its contribution is negligible compared to AN, VOL and OCN.

Done. See P13L856-857.

---

## Editor Decision (ED1)

Editor comments.
Line numbers refer to the manuscript without track changes.
* * *
Abstract: Your abstract has ~340 words. Please pay attention to the ACP author guidelines that recommend a length of 250 words. I realize that the abstract got longer due to the revision; however, some of the text seems unnecessarily wordy, e.g.

*There are much larger DMS concentrations simulated close to the sea surface than observed, indicating that the DMS emissions may be too high from all models*

could be shortened to

*Simulated DMS concentrations near the sea surface exceed observed levels, suggesting potential overestimation of DMS emissions in all models.*

or

*In this study, we compare the spatial and temporal distribution of four sulfur-containing species, dimethyl sulfide (DMS), sulfur dioxide (SO2), particulate methanesulfonate (MSA), and particulate sulfate (SO4), that were measured during the airborne NASA Atmospheric Tomography (ATom) mission and simulated by five AeroCom-III models to analyze the budget of sulfur cycle from the models.*

could be shortened to

*We compare spatially and temporally resolved measurements from the NASA ATom mission with simulations from five AeroCom-III models for four species of the sulfur cycle (dimethyl sulfide (DMS), sulfur dioxide (SO2), particulate methanesulfonate (MSA), and particulate sulfate (SO4)).*

*Etc*

Please try to shorten the abstract accordingly.

Minor/technical comments:

l. 72: replace 'sun's rays' with 'solar radiation'

l. 73: replace 'affect cloud physics' by 'act as efficient cloud condensation nucleus'

l. 79: replace 'radiation forcing' by 'radiative forcing'

l. 80: replace 'model results' by 'model studies'

l. 126/7: This sentence seems somewhat out of place or misleading. The sulfur cycle in the ocean likely differs a lot from that in the atmosphere, involving other compounds (e.g. H2S, thionates), concentrations and fluxes. Please clarify why the oceanic sulfur cycle is relevant.

l. 159/60: Can this informaiton be simply added to the sentence in line 154, e.g. '...high-resolution time-of-flight aerosol mass spectrometer at 1-s and 60-s resolution'?

l. 214: 'for all AToms' sounds colloquial. Do you mean 'for all ATom periods'?

l. 243: 'computational efficiency of scientific research' sounds odd. Do you mean 'computational efficiency of the aerosol models'?

Figures 2 and 3 (and wherever else applicable):
(i) Spell ATom consistently with the text (i.e. upper case 'AT' and lower case 'om')
(ii) specify 'statistical values' in the caption – I assume that you mean 'median' and 'mean' (please also check the manuscript and use the more accurate terms where appropriate e.g. l. 298, 311)

Figures S3 and S4: Adding so many numbers in the figure panels and even overlapping the traces makes the figure look messy. I suggest that you create a table comparing the median and mean values of Figures 2, 3, S3, S4. This way the reader could also more clearly see the 'substantial drop' as referred to as in l. 317.

l. 313: what do you mean by 'unobserved episode events'? Is it important to mention or could it be deleted?

l. 317/8: 'but the model statistics change relatively small' is not clear.

l. 334: This sentence should be simplified or split into two, e.g. (please check whether this reflects the intended meaning!)

The median of the predicted DMS concentrations is higher in most cases than that of the observations. The difference between the modeled and observed mean values is much higher (more than a factor of 10) than that of the median values. This reflects a few very high predicted DMS values.

l. 361: replace 'span' by 'range'

l. 363: remove 'easily'.

l. 368: 'E3SM performed SO4 simulations among other models.' is misleading and/or colloquial. Do you mean 'the results of E3SM are generally within the ranges as predicted by the other models'?

l. 372: 'The complexity of  the chemistry deserves more attention' – this is a very generic vague sentence. Either specify what you want to say or remove.

l. 380: Do you mean 'than by the other two AeroCom models'?

l. 386: do you mean indeed 'provides' or 'predicts'?

l. 409: do you mean 'the data set is large' or 'the values are large'?

l. 411 and l. 412: words seem missing here, e.g., 'inventory' or 'data base', since the inter/extrapolation techniques did not create emissions.

l. 415/6: What do you want to say here? CAM-ATRAS and OsloCTM3 cannot 'report anything in Section 4'. Do you mean that CAM-ATRAS and OsloCTM3 predicted similar values for DMS emissions as discussed in Section 4?

l. 423: Data can be collected in the atmosphere but not by models. What do you mean here?

l. 430: 'to a sulfur compound' can be removed.

l. 437: 'Of the five models…' seems to be redundant and repeating information of line 425.

l. 440: What do you mean by 'phase stages'? Would 'gas-particle partitioning' be more accurate?

l. 451: What do you mean by 'anomalous behavior' (what woudl be normal behavior)? Can you use the word 'bias' here?

l. 452: which knowledge? Be more accurate here.

l. 458: 'The most high concentration areas' – do you mean 'the areas with the highest concentrations' or 'most of the areas that show high concentration'?

l. 459: 'Things are a bit more complicated' is very colloquial – rephrase

l. 464: 'tropospheric atmosphere' should  be replaced by 'troposphere'

l. 465: 'in the Pacific is higher than that in the Atlantic' would imply $SO_4$ concentrations in the oceans – I assume that you refer here to 'above the Pacific and Atlantic oceans'

l. 468: 'simulated and observed worlds' should be replaced by 'simulations and observations'

l. 469: 'Differences may be caused by majority models or a few individual models.' What do you mean here?

l. 481: What do you mean by  'potential source of horizontal transport' ? Horizontal transport is a consequence of dynamics, i.e. advection. Or do you mean 'potential source of $SO_2$ by horizontal transport'?

l. 510: replace 'reserves' by 'serves as'

l. 525: replace 'has' by 'predicts' (GEOS predicts…)

l. 528: 'models CAM-ATRAS and OsloCTM3 emit highest DMS' – please improve colloquial wording. Models do not emit anything.

l. 532: replace 'reversely' by 'inversely'

l. 534: why is 'diversity' in quotation marks? What is meant here?

l. 547: do you mean 'predictor' rather than 'simulation' here?

l. 578: is 'and' redundant?

l. 586/7: Should 'its' be replaced by 'their' (i.e. related to the oceanic sources)?

l. 599: 'but' seems redundant here

l. 603: 'orchid' is not a common color. Either call it 'pink' or remove (there is only one shaded area in the figure)

l. 625: Be more specific: replace 'sulfur' by 'sulfur species' or 'sulfur cycle'

l. 634: remove 'sources' and 'resultant'

l. 669: 'Several potential directions for improving sulfur simulations are suggested above.' – Either remove this sentence or specify where 'above' and which directions.

l. 671: 'all modelers should focus on the calculation of the air-sea exchange flux formula, as it plays a key role in determining DMS emissions.' – this is very vague. Certainly not 'all modelers' ... be clearer and more accurate here. What type of models are you referring to?

l. 677: This is a very vague concluding sentence of your study. I suggest removing it.

---

## Author Response (AR2)

Thanks to the editor for the good suggestions and careful editing.

Editor comments.
Line numbers refer to the manuscript without track changes.
- - - - - - - - - - - - - - - - - - - - - - -

Abstract: Your abstract has ~340 words. Please pay attention to the ACP author guidelines that recommend a length of 250 words. I realize that the abstract got longer due to the revision; however, some of the text seems unnecessarily wordy, e.g.

*There are much larger DMS concentrations simulated close to the sea surface than observed, indicating that the DMS emissions may be too high from all models*

could be shortened to

*Simulated DMS concentrations near the sea surface exceed observed levels, suggesting potential overestimation of DMS emissions in all models.*

or

*In this study, we compare the spatial and temporal distribution of four sulfur-containing species, dimethyl sulfide (DMS), sulfur dioxide (SO2), particulate methanesulfonate (MSA), and particulate sulfate (SO4), that were measured during the airborne NASA Atmospheric Tomography (ATom) mission and*
*simulated by five AeroCom-III models to analyze the budget of sulfur cycle from the models.*

could be shortened to

*We compare spatially and temporally resolved measurements from the NASA ATom mission with simulations from five AeroCom-III models for four species of the sulfur cycle (dimethyl sulfide (DMS), sulfur dioxide (SO2), particulate methanesulfonate (MSA), and particulate sulfate (SO4)).*

*Etc*

Please try to shorten the abstract accordingly.

Thanks. We adopted the suggestions and revised the abstract. There are now 247 words.

Minor/technical comments:

l. 72: replace 'sun's rays' with 'solar radiation'
Done

l. 73: replace 'affect cloud physics' by 'act as efficient cloud condensation nucleus'
Done

l. 79: replace 'radiation forcing' by 'radiative forcing'
Done

l. 80: replace 'model results' by 'model studies'

Done

l. 126/7: This sentence seems somewhat out of place or misleading. The sulfur cycle in the ocean likely differs a lot from that in the atmosphere, involving other compounds (e.g. H2S, thionates), concentrations and fluxes. Please clarify why the oceanic sulfur cycle is relevant.
Change "in the ocean" to "…. over the remote ocean".

l. 159/60: Can this informaiton be simply added to the sentence in line 154, e.g. '...high-resolution time- of-flight aerosol mass spectrometer at 1-s and 60-s resolution'?
We thought through the logical flow. It seems best to leave this sentence where it is. This sentence is logically related to the next three sentences, which discuss detection limits and negative measurements associated with different data averaging intervals.

l. 214: 'for all AToms' sounds colloquial. Do you mean 'for all ATom periods'?
Change it to "for all ATom periods".

l. 243: 'computational efficiency of scientific research' sounds odd. Do you mean 'computational efficiency of the aerosol models'?
Change it to "computational efficiency of the aerosol models".

Figures 2 and 3 (and wherever else applicable):
(i) Spell ATom consistently with the text (i.e. upper case 'AT' and lower case 'om')
    Done
(ii) specify 'statistical values' in the caption – I assume that you mean 'median' and 'mean' (please also check the manuscript and use the more accurate terms where appropriate e.g. l. 298, 311)
In Fig. 2 cation, we define the statistical values as "Statistical values include the range of the data from minimum to maximum, the three levels of the 25$^{th}$, 50$^{th}$ (median), and 75$^{th}$ percentiles in the box, and the filled circle for the mean." We now place this sentence before the sentence that refers to it.

Figures S3 and S4: Adding so many numbers in the figure panels and even overlapping the traces makes the figure look messy. I suggest that you create a table comparing the median and mean values of Figures 2, 3, S3, S4. This way the reader could also more clearly see the 'substantial drop' as referred to as in l. 317.
Done. We now have Table S1 to give these numbers for Figs. 2-4 and Table S2 for Figs. S2-4. Numbers on Figs. S2-4 have been removed.

l. 313: what do you mean by 'unobserved episode events'? Is it important to mention or could it be deleted?
Change the sentence to "…. episode events that were not reported in measurements."

l. 317/8: 'but the model statistics change relatively small' is not clear.
Delete "but the model statistics change relatively small".

l. 334: This sentence should be simplified or split into two, e.g. (please check whether this reflects the intended meaning!)

The median of the predicted DMS concentrations is higher in most cases than that of the observations.

The difference between the modeled and observed mean values is much higher (more than a factor of 10) than that of the median values. This reflects a few very high predicted DMS values.

Change the sentence to "The model DMS median values are mostly higher than the observed values. The model DMS mean values are even higher than the observed means (sometimes by more than a factor of 10). This reflects a few very high predicted DMS values."

l. 361: replace 'span' by 'range'
Done.

l. 363: remove 'easily'.
Done.

l. 368: 'E3SM performed SO4 simulations among other models.' is misleading and/or colloquial. Do you mean 'the results of E3SM are generally within the ranges as predicted by the other models'?
Yes. The sentence has been revised accordingly.

l. 372: 'The complexity of the chemistry deserves more attention' – this is a very generic vague sentence. Either specify what you want to say or remove.
Remove it.

l. 380: Do you mean 'than by the other two AeroCom models'?
Change the sentence to "SO$_2$ is better simulated by model IMPACT in the NH than other four AeroCom models and by models CAM-ATRAS and OsloCTM3 in the SH than other three AeroCom models."

l. 386: do you mean indeed 'provides' or 'predicts'?
"Predicts" seems more relevant here. Done.

l. 409: do you mean 'the data set is large' or 'the values are large'?
Change the sentence to "… the data set is large …'.

l. 411 and l. 412: words seem missing here, e.g., 'inventory' or 'data base', since the inter/extrapolation techniques did not create emissions.
Change the sentence to "… the data set …'. Also see answer to l. 409.

l. 415/6: What do you want to say here? CAM-ATRAS and OsloCTM3 cannot 'report anything in Section 4'. Do you mean that CAM-ATRAS and OsloCTM3 predicted similar values for DMS emissions as discussed in Section 4?
Change Section 4 to Table 4. CAM-ATRAS and OsloCTM3 predicted similar values for DMS emissions in Table 4 (and discussed in Section 4).

l. 423: Data can be collected in the atmosphere but not by models. What do you mean here?
Change the sentence to "The data submitted by the AeroCom models …".

l. 430: 'to a sulfur compound' can be removed.
Done.

l. 437: 'Of the five models...' seems to be redundant and repeating information of line 425.

Remove the sentence.

l. 440: What do you mean by 'phase stages'? Would 'gas-particle partitioning' be more accurate?
Change as suggested.

l. 451: What do you mean by 'anomalous behavior' (what woudl be normal behavior)? Can you use the word 'bias' here?
Change as suggested.

l. 452: which knowledge? Be more accurate here.
Change the sentence to "With the information provided by Figs. S9-11, …".

l. 458: 'The most high concentration areas' – do you mean 'the areas with the highest concentrations' or 'most of the areas that show high concentration'?
Change the sentence to "The areas with the highest concentrations ….".

l. 459: 'Things are a bit more complicated' is very colloquial – rephrase
Remove it.

l. 464: 'tropospheric atmosphere' should be replaced by 'troposphere'
Done.

l. 465: 'in the Pacific is higher than that in the Atlantic' would imply SO4 concentrations in the oceans – I assume that you refer here to 'above the Pacific and Atlantic oceans'
Yes. Change the sentence to "…. over the Pacific is higher than that over the Atlantic ….'.

l. 468: 'simulated and observed worlds' should be replaced by 'simulations and observations'
Done.

l. 469: 'Differences may be caused by majority models or a few individual models.' What do you mean here?
Figs.10-12 show the difference between observed and simulated values for different regions and ATom periods. The simulated values are the averages of multi-model median (MMM). Bias in the MMM can be attributed to the majority of participating AeroCom models or to individual models, depending on the where and when the comparison is made. Examples explaining this are given in the discussions that follow this sentence.

l. 481: What do you mean by 'potential source of horizontal transport' ? Horizontal transport is a consequence of dynamics, i.e. advection. Or do you mean 'potential source of SO2 by horizontal transport'?
Change to "potential source of SO2 by horizontal transport".

l. 510: replace 'reserves' by 'serves as'
Done.

l. 525: replace 'has' by 'predicts' (GEOS predicts...)
Done.

l. 528: 'models CAM-ATRAS and OsloCTM3 emit highest DMS' – please improve colloquial wording. Models do not emit anything.
Change the sentence to "…, models CAM-ATRAS and OsloCTM3 predict the highest DMS emissions …'.

l. 532: replace 'reversely' by 'inversely'
Done.

l. 534: why is 'diversity' in quotation marks? What is meant here?
Remove quotation marks here.

l. 547: do you mean 'predictor' rather than 'simulation' here?
Change to "predictor".

l. 578: is 'and' redundant?
Remove "and".

l. 586/7: Should 'its' be replaced by 'their' (i.e. related to the oceanic sources)?
Yes. Change to "their".

l. 599: 'but' seems redundant here
Replace "but" with ",".

l. 603: 'orchid' is not a common color. Either call it 'pink' or remove (there is only one shaded area in the figure)
Remove "orchid".

l. 625: Be more specific: replace 'sulfur' by 'sulfur species' or 'sulfur cycle'
Change to "sulfur species".

l. 634: remove 'sources' and 'resultant'
Done.

l. 669: 'Several potential directions for improving sulfur simulations are suggested above.' – Either remove this sentence or specify where 'above' and which directions.
Remove this sentence here and add "to improve sulfur simulation" in l. 755 in revised (R2) highlight version.

l. 671: 'all modelers should focus on the calculation of the air-sea exchange flux formula, as it plays a key role in determining DMS emissions.' – this is very vague. Certainly not 'all modelers' ... be clearer and more accurate here. What type of models are you referring to?
Change sentence to "All modelers involved in this work should focus on the calculation of the air-sea exchange flux formula as it plays a key role in determining DMS emissions. To our knowledge, many other aerosol models employ similar formulas in air-sea flux calculations, so the findings here are applicable to them as well."

l. 677: This is a very vague concluding sentence of your study. I suggest removing it.
Done.